# Politicization of COVID-19 health-protective behaviors in the United States: Longitudinal and cross-national evidence

Wolfgang Stroebe[1], Michelle R. vanDellen[2]*, Georgios Abakoumkin[3], Edward P. Lemay, Jr.[4], William M. Schiavone[2], Maximilian Agostini[1], Jocelyn J. Bélanger[5], Ben Gützkow[1], Jannis Kreienkamp[1], Anne Margit Reitsema[1], Jamilah Hanum Abdul Khaiyom[6‡], Vjolica Ahmedi[7‡], Handan Akkas[8‡], Carlos A. Almenara[9‡], Mohsin Atta[10‡], Sabahat Cigdem Bagci[11‡], Sima Basel[5‡], Edona Berisha Kida[7‡], Allan B. I. Bernardo[12‡], Nicholas R. Buttrick[13‡], Phatthanakit Chobthamkit[14‡], Hoon-Seok Choi[15‡], Mioara Cristea[16‡], Sára Csaba[17‡], Kaja Damnjanović[18‡], Ivan Danyliuk[19‡], Arobindu Dash[20‡], Daniela Di Santo[21‡], Karen M. Douglas[22‡], Violeta Enea[23‡], Daiane Gracieli Faller[5‡], Gavan Fitzsimons[24], Alexandra Gheorghiu[23], Ángel Gómez[25‡], Ali Hamaidia[26‡], Qing Han[27‡], Mai Helmy[28‡], Joevarian Hudiyana[29‡], Bertus F. Jeronimus[1‡], Ding-Yu Jiang[30‡], Veljko Jovanović[31‡], Željka Kamenov[32‡], Anna Kende[17‡], Shian-Ling Keng[33‡], Tra Thi Thanh Kieu[34‡], Yasin Koc[1‡], Kamila Kovyazina[35‡], Inna Kozytska[10‡], Joshua Krause[1‡], Arie W. Kruglanksi[4‡], Anton Kurapov[19‡], Maja Kutlaca[36‡], Nóra Anna Lantos[17‡], Cokorda Bagus Jaya Lemsmana[37‡], Winnifred R. Louis[38‡], Adrian Lueders[39‡], Najma Iqbal Malik[10‡], Anton Martinez[40‡], Kira O. McCabe[41‡], Jasmina Mehulić[32‡], Mirra Noor Milla[29‡], Idris Mohammed[42‡], Erica Molinario[43‡], Manuel Moyano[44‡], Hayat Muhammad[45‡], Silvana Mula[21‡], Hamdi Muluk[29‡], Solomiia Myroniuk[1‡], Reza Najafi[46‡], Claudia F. Nisa[5‡], Boglárka Nyúl[17‡], Paul A. O'Keefe[33‡], Jose Javier Olivas Osuna[47‡], Evgeny N. Osin[48‡], Joonha Park[49‡], Gennaro Pica[50‡], Antonio Pierro[21‡], Jonas Rees[51‡], Elena Resta[21‡], Marika Rullo[52‡], Michelle K. Ryan[53,1‡], Adil Samekin[54‡], Pekka Santtila[55‡], Edyta Sasin[5‡], Birga M. Schumpe[56‡], Heyla A. Selim[57‡], Michael Vicente Stanton[58‡], Samiah Sultana[1‡], Robbie M. Sutton[22‡], Eleftheria Tseliou[3‡], Akira Utsugi[59‡], Jolien Anne van Breen[60], Caspar J. Van Lissa[61‡], Kees Van Veen[1‡], Alexandra Vázquez[25‡], Robin Wollast[39‡], Victoria Wai-Lan Yeung[62‡], Somayeh Zand[46‡], Iris Lav Žeželj[18‡], Bang Zheng[63‡], Andreas Zick[51‡], Claudia Zúñiga[64‡], N. Pontus Leander[1]

1 Department of Psychology, University of Groningen, Groningen, Netherlands, 2 Department of Psychology, University of Georgia, Athens, Georgia, United States of America, 3 Laboratory of Psychology, Department of Early Childhood Education, University of Thessaly, Volos, Greece, 4 Department of Psychology, University of Maryland, College Park, Maryland, United States of America, 5 Department of Psychology, New York University Abu Dhabi, Abu Dhabi, United Arab Emirates, 6 Department of Psychology, International Islamic University Malaysia, Selangor, Malaysia, 7 Faculty of Education, Pristine University, Pristina, Kosovo, 8 Organizational Behavior, Ankara Science University, Ankara, Turkey, 9 Faculty of Health Science, Universidad Peruana de Ciencias Aplicadas, Lima, Peru, 10 Department of Psychology, University of Sargodha, Sargodha, Pakistan, 11 Department of Psychology, Sabanci University, Tuzla, Turkey, 12 Department of Psychology, De La Salle University, Manila, Philippines, 13 Department of Psychology, University of Virginia, Charlottesville, Virginia, United States of America, 14 Department of Psychology, Thammasat University, Bangkok, Thailand, 15 Department of Psychology, Sungkyunkwan University, Seoul, South Korea, 16 Department of Psychology, Heriot Watt University, Edinburgh, Scotland, 17 Department of Psychology, ELTE Eötvös Loránd University, Budapest, Hungary, 18 Department of Psychology, University of Belgrade, Belgrade, Serbia, 19 Department of Psychology, Taras Shevchenko National University of Kyiv, Kyiv, Ukraine, 20 Department of Social Sciences, International University of Business Agriculture and Technology, Dhaka, Bangladesh, 21 Department of Social and Developmental Psychology, University "La Sapienza", Rome, Italy, 22 School of Psychology, University of Kent, Kent, United Kingdom, 23 Department of Psychology, Alexandru Ioan Cuza University, Iași, Romania, 24 Departments of Marketing and Psychology, Duke University, Durham, North Carolina, United States of America, 25 Center for European Studies, Faculty of Law, Universidad Nacional de Educacion a Distancia, Madrid, Spain, 26 Department of Psychology and Human Resources Development, Setif 2 University, Setif, Algeria, 27 The School of Psychological Science, University of Bristol, Bristol, United Kingdom, 28 Department of Psychology,



**Data Availability Statement:** Data cannot be shared publicly because the institution governing the data collection and management has deemed political orientation as a sensitive personal piece of

data. Data are available for the data managers of the PsyCorona project for researchers who meet the criteria for access to confidential data. This contact should go to the psycorona@rug.nl email address. The authors had no special access privileges to the data that others requesting the data will not have.

**Funding:** This research received support from the New York University Abu Dhabi (VCDSF/75-71015) to J.N., the University of Groningen (Sustainable Society & Ubbo Emmius Fund) to N.P.L., and the Instituto de Salud Carlos III (COV20/00086) co-funded by the European Regional Development Fund (ERDF 'A way to make Europe' to M.M. The funders had no role in study design, data collection and analysis, decision to publish, or preparation of the manuscript.

**Competing interests:** The authors have declared that no competing interests exist.

Menoufia University, Al Minufiyah, Egypt, **29** Department of Psychology, Universitas Indonesia, Jakarta, Indonesia, **30** Department of Psychology, National Chung-Cheng University, Minxiong, Taiwan, **31** Department of Psychology, University of Novi Sad, Novi Sad, Serbia, **32** Faculty of Humanities and Social Sciences, University of Zagreb, Zagreb, Croatia, **33** Division of Social Science, Yale-NUS College, Singapore, Singapore, **34** Department of Psychology, HCMC University of Education, Ho Chi Minh City, Vietnam, **35** Independent Researcher, Kazakhstan, **36** Department of Psychology, Durham University, Durham, United Kingdom, **37** Department of Psychiatry, Udayana University, Bali, Indonesia, **38** School of Psychology, The University of Queensland, Brisbane, Australia, **39** Laboratoire de Psychologie Sociale et Cognitive, Université Clermont-Auvergne, Clermont-Ferrand, France, **40** Department of Psychology, University of Sheffield, Sheffield, United Kingdom, **41** Department of Psychology, Carleton University, Ottawa, Canada, **42** Mass Communication, Usmanu Danfodiyo University Sokoto, Sokoto, Nigeria, **43** Department of Psychology, Florida Gulf Coast University, Fort Myers, Florida, United States of America, **44** Department of Psychology, University of Cordoba, Andalusia, Spain, **45** Department of Psychology, University of Peshawar, Peshawar, Pakistan, **46** Department of Psychology, Islamic Azad University, Rasht, Iran, **47** Department of Political Science and Administration, National Distance Education University (UNED), Madrid, Spain, **48** Department of Psychology, HSE University, Moscow, Russia, **49** Graduate School of Management, NUCB Business School, Nagoya, Japan, **50** School of Law, University of Camerino, Camerino, Italy, **51** Research Institute Social Cohesion, Institute for Interdisciplinary Research on Conflict and Violence, and Department of Social Psychology Bielefeld University, Bielefeld, Germany, **52** Department of Educational, Humanities and Intercultural Communication, University of Siena, Siena, Italy, **53** Department of Psychology, University of Exeter, Exeter, United Kingdom, **54** School of Liberal Arts, M. Narikbayec KAZGUU University, Nur-Sultan, Kazakhstan, **55** Department of Psychology, New York University Shanghai, Shanghai, China, **56** Faculty of Social and Behavioural Sciences, University of Amsterdam, Amsterdam, Netherlands, **57** Department of Psychology, King Saud University, Riyadh, Saudi Arabia, **58** Department of Public Health, California State University East Bay, Hayward, California, United States of America, **59** Department of Psychology, Nagoya University, Nagoya, Japan, **60** Institute of Governance and Global Affairs, Leiden University, Leiden, Netherlands, **61** Department of Methodology & Statistics, Utrecht University, Utrecht, Netherlands, **62** Department of Psychology, Lingnan University, Hong Kong, China, **63** Ageing Epidemiology Research Unit, School of Public Health, Faculty of Medicine, Imperial College London, London, United Kingdom, **64** Department of Psychology, Universidad de Chile, Santiago de Chile, Chile

☯ These authors contributed equally to this work.
‡ These authors also contributed equally to this work.
\* mvd@uga.edu

## Abstract

During the initial phase of the COVID-19 pandemic, U.S. conservative politicians and the media downplayed the risk of both contracting COVID-19 and the effectiveness of recommended health behaviors. Health behavior theories suggest perceived vulnerability to a health threat and perceived effectiveness of recommended health-protective behaviors determine motivation to follow recommendations. Accordingly, we predicted that—as a result of politicization of the pandemic—politically conservative Americans would be less likely to enact recommended health-protective behaviors. In two longitudinal studies of U.S. residents, political conservatism was inversely associated with perceived health risk and adoption of health-protective behaviors over time. The effects of political orientation on health-protective behaviors were mediated by perceived risk of infection, perceived severity of infection, and perceived effectiveness of the health-protective behaviors. In a global cross-national analysis, effects were stronger in the U.S. ($N$ = 10,923) than in an international sample (total $N$ = 51,986), highlighting the increased and overt politicization of health behaviors in the U.S.

## Introduction

Prior to the development of vaccines, behavioral measures were the primary means of preventing the spread of COVID-19. The World Health Organization (WHO) and the U.S. Centers for Disease Control (CDC) recommended a number of health-protective behaviors to lower a person's risk of contracting and spreading the virus. The initial list of such recommendations included hand washing, social distancing, and self-quarantining, followed by an additional recommendation to wear face masks and face coverings. The effectiveness of lockdowns imposed in multiple countries demonstrated the potential of extreme social distancing to prevent infection [1, 2]. However, considering the severe economic consequences of countrywide lockdowns, many countries relied on individual decision-making to contain the spread of COVID-19. With the availability of vaccines in 2021, these countries are relying on the willingness of individuals (including essential care workers in healthcare, education and other high contact fields) to be inoculated. A central question, therefore, is whether individuals' willingness to adopt health-protective behaviors and to be vaccinated varies with their subjective perceptions about COVID-19, perceptions that may be shaped by political concerns and politicized social influence.

According to theories of health behavior, individuals' compliance with recommendations depends on their perceptions of infection risk, the anticipated severity of such an infection, and the perceived effectiveness of recommended health-protective behaviors [3–9]. For example, the Health Belief Model, a widely tested theory of health behavior, asserts that the likelihood of individuals engaging in a given health-protective behavior is determined by the *perceived severity* of the health threat and the *perceived effectiveness* of the recommended health-protective behavior [5, 9]. The perceived severity of a health threat is determined by the extent to which individuals believe they are likely to contract an illness and how severe they anticipate the personal consequences of that illness to be. Irrespective of this perception, however, the likelihood of individuals engaging in a recommended health-protective behavior will depend on whether they perceive the recommended measure to be *effective* in preventing the health threat and whether the perceived benefits of that behavior outweigh the perceived costs [5, 9, 10]. Other health behavior theories—such as the Protection Motivation Theory—confirm the importance of these perceptions for the adoption of health-protective behavior [7].

In accordance with these theories, individuals would be expected to adopt health-protective behaviors to prevent a COVID-19 infection to the extent they believe they could become infected and consider such an infection to be a serious threat to their health. Whether or not people adopt these recommended health-protective behaviors would also be influenced by the perceived effectiveness of that behavior in preventing an infection. Two large meta-analyses on the effectiveness of fear-arousing communications have provided empirical support for the role of perceived threat and perceived behavioral effectiveness in predicting health behaviors from both experimental and observational studies [4, 9].

## Political beliefs are associated with differing perceptions of health risks

Several studies provide evidence for a relationship between individual-level political orientation and perception of health risk associated with COVID-19, as well as compliance with recommended health-protective behaviors in the U.S. [11–20]. In the U.S., early public polls indicated partisan differences in perceptions of health threat posed by COVID-19 [21, 22]. Geotracking data of 15 million smartphones suggested that people who lived in counties that voted for Trump in the 2016 U.S. election were 14% less likely to engage in recommended social distancing behaviors [16]. Another study based on the daily reported activities data of more than a million Americans indicated that political partisanship predicted reduced physical and social mobility much more strongly than did the local incidence of COVID-19 [13].

For people living in the U.S., perceptions of both the threat of being infected with COVID-19 and the effectiveness of the recommended health-protective behaviors are possible explanations for these political differences in compliance with recommended health-protective behaviors. What remains unclear, however, is whether such political differences in behavior merely reflect consistent differences in the impact of political ideologies on behavior, or whether they reflect dynamic, politicized forms of social influence. Although conservative-leaning Americans generally perceive their environments as more threatening [23, 24], they deemphasized the public health threat, instead focusing on perceived threats to the economy and personal liberty that would result from pandemic-related preventive measures. Such a shift is reminiscent of studies on solution aversion which showed people deny the existence of a problem when presented with a solution they perceive as politically unpalatable (such as with cap-and-trade or gun control; Kay & Campbell, 2014 [25]).

Politicized social influence may be exercised and maintained through partisan messaging and information consumption. Public communication from the right-leaning Trump White House consistently downplayed the seriousness of COVID-19 and the risk of getting infected. For instance, on February 26[th] of 2020, the President publicly called the coronavirus "a regular flu", stated there were few cases in the U.S, and that the pandemic was under control [26]. Similar statements were made by right-leaning politicians with regards to the efficacy of mask-wearing and social distancing recommendations [27, 28], with some of them hosting indoor and maskless election rallies that defied state regulations and CDC recommendations [29]. These behaviors convey the message that the recommended health-protective behaviors are neither necessary nor effective.

Liberal (economically left-leaning) Americans and conservative (right-leaning) Americans Liberal (economically left-leaning) Americans and conservative (economically right-leaning) Americans tend to rely on different sources of information, which prioritize different values. Perceptions of the credibility of these sources also vary as a function of political orientation [30]. The credibility of a source can be an important determinant of the impact of communication [31, 32], particularly if respondents' motivation and ability to scrutinize the communication is low [33]. If conservatives believed their vulnerability to a health threat to be low, they would be less motivated to carefully scrutinize health communication [3, 4] and would therefore be more likely to accept information from a source they consider credible [33].

According to a survey by the PEW Research Center [30], 76% of liberals said that the CDC and other public health experts *"get the facts right almost all of the time"* with regards to the COVID-19 outbreak, whereas only 51% of conservatives agreed with this statement. In contrast, 54% of conservatives believed that the Trump White House got its facts right compared to 9% of liberals. Differences in the information sources relied on and trusted by conservatives and liberals may have exacerbated perceptions of the seriousness of the COVID-19 pandemic. An academic study based on a representative sample of Americans taken in March 2020 similarly found that liberals place less trust in politicians to handle the pandemic and are more trusting of medical experts such as the WHO [19]. Similarly, results from a representative survey of Americans adults administered in September 2020 indicated that trust that the World Health Organization is capable of effectively managing the pandemic and providing reliable information about COVID-19 is predicted by Democratic Party identity, liberal ideology, and a strong internationalist foreign policy orientation. Trust in the competence of the WHO is also a strong predictor of both social distancing and compliance with COVID-19 guidelines. However, this effect is reduced when trust in the CDC is also taken into account. Finally, a study found that Americans, who identified as Republicans or Independents perceived a COVID infection as less severe, were less fearful of getting infected, had less knowledge about COVID-19, had less trust in science and were less prepared to comply with health behavior

recommendations [14]. In summary then, compared to liberals, conservatives are less likely to trust science and the information provided by scientific organization such as the CDC and the WHO and rather rely on information provided by politicians of their own political persuasion. As a result, they are less informed about the pandemic, are less fearful of getting infected and are also less prepared to comply with the health recommendations.

## The present research

In the context of COVID-19, threat perceptions and associated health-protective behaviors are disproportionately adopted by liberals compared to their conservative counterparts. To the extent that this effect is localized to the U.S., it would further suggest the effect of politicized social influence, as opposed to ideological differences between conservative and liberal ideologies. The two studies reported in this paper apply a health psychological model–the Health Belief Model—to a social psychological problem, namely the association between political orientation and people's acceptance of and compliance with recommendations for health-protective behaviors. The starting point of our studies is the well-documented assertion that the Trump White House and conservative-leaning information sources systematically deemphasized the seriousness of COVID-19 and the effectiveness of the WHO and CDC recommendations regarding health-protective behaviors. To the extent that political orientation reflects differences in COVID-19 information consumption patterns, we expected that conservatives would perceive both the risk of becoming personally infected and the protective effects of health behaviors as lower. As a consequence, they would be less motivated to engage in recommended health-protective behaviors. Most importantly, we further predicted that any politicized adoption of health-protective behaviors would be mediated by political differences in the *perceived risk* of contracting the virus, the *perceived severity* of the consequences of such an infection, and the *perceived effectiveness* of the recommended health-protective behaviors.

We tested these hypotheses in two studies with samples of participants living in the U.S. The second study also included an international sample for comparison. In both studies, we assessed political orientation (conservative vs. liberal), perceived risk of getting infected, and willingness to engage in recommended health-protective behaviors. In Study 2, we additionally assessed perceived severity of getting infected and the perceived effectiveness of wearing a face covering. In both studies, participants were resampled for several weeks, allowing for examination of the effects across time (5 time points in Study 1 and 13 time points in Study 2). Finally, Study 2 also allowed for a comparison of the relationship between politics and health-protective behavior in the U.S. relative to other countries. This comparison would enable us to rule out the possibility that the association between political orientation and virus perception in the U.S. could merely be the result of different worldviews, or beliefs inherent to conservative and liberal ideologies. We hypothesized these effects would be stronger in the U.S. compared to other countries. Support for this prediction would suggest the effects of political conservatism on lower risk perceptions and health-protective behaviors are due to sources of influence that are localized to the U.S.

## Study 1

### Method

**Participants and procedure.** This study involved longitudinally tracking participants' attitudes and self-reported behaviors across five time points. Wave 1 (Baseline) was launched on March 10th, one day before the WHO declared the COVID-19 outbreak a pandemic. To capture potentially acute and relatively long-term changes, we followed up with participants at

three time points in close succession (March 20[th], March 28[th], and April 11[th], 2020) as well as a longer-term follow-up on June 16[th], 2020.

Participants were Amazon MTurk respondents. They were recruited to "fill out five surveys across the next months asking questions about recent events in society." Current residence in the U.S. was an eligibility criterion, and we used an IP address filter to ensure fulfillment of this requirement. At Baseline, 1,056 MTurk respondents participated in the study. Seventeen individuals were excluded from analyses due to suspicion of data invalidity (e.g., double MTurk ID; survey completion in less than five minutes); thus, the final sample size was $N = 1,039$. Table 1 reports characteristics of these participants. At Wave 2, 649 participants yielded valid data (data from seven individuals were excluded), at Wave 3, there were 642 participants with valid data (seven individuals were excluded), there were 547 participants at Wave 4 (nine were excluded), and 462 participated in Wave 5 (one was excluded). Effect sizes were not anticipated prior to data collection.

**Measures.** A critical aim of this study was to capture attitudes and behaviors as quickly during the pandemic as possible. Our approach was to select and use brief face-valid measures. This decision afforded high response rates to surveys, allowed available funds to be used to expand the sample size, and ultimately afforded the translation of items into 30 languages in Study 2. Moreover, short measures are not faulty per se, but can be psychometrically appropriate [34–36].

*Perceived risk of infection.* Perceived risk of infection was assessed at all five time points with an adapted threat likelihood item adapted [37]: "How likely is it that the following will happen to you in the next few months? . . . You will get infected with the Coronavirus." (1 = *Not at all likely*; 5 = *Extremely likely*).

*Health-protective behaviors.* In this study, we assessed health-protective behaviors based on the three recommendations made by the WHO. At the start of this study, the health-protective

**Table 1. Demographic information at baseline for participants in Studies 1 and 2.**

|  | Study 1 (U.S.) | Study 2 (U.S.) | Study 2 (Non-U.S.) |
|---|---|---|---|
|  | *N* | *N* | *N* |
| **Gender** |  |  |  |
| Male | 463 | 4043 | 19732 |
| Female | 529 | 6773 | 31704 |
| Other | 6 | 81 | 223 |
| Did not report | 41 | 26 | 327 |
| **Age** |  |  |  |
| 18–24 | 62 | 1670 | 12746 |
| 25–34 | 367 | 3244 | 11991 |
| 35–44 | 256 | 2446 | 9554 |
| 45–54 | 153 | 1534 | 7518 |
| 55–64 | 111 | 1211 | 5739 |
| 65+ | 49 | 784 | 4086 |
| Did not report | 41 | 34 | 352 |
| **Education** |  |  |  |
| Some High School or less | 7 | 360 | 547 |
| High School graduate/GED | 85 | 1637 | 12601 |
| Some College | 211 | 2146 | 12549 |
| College Graduate | 415 | 4229 | 14834 |
| Graduate Degree | 261 | 2512 | 11044 |
| Did not report | 60 | 39 | 411 |

**Table 2. Relationship of baseline political orientation with perceived health risk and health-protective behaviors: Study 1.**

| Date | Perceived health risk | | WHO Health-protective behaviors | |
|---|---|---|---|---|
| | *M (SD), N* | *r (N)* | *M (SD), N* | *r (N)* |
| March 10[th] | 2.55 (1.13), 1029 | .138 (1001) | 1.84 (1.04), 1021 | .093 (1001) |
| March 20[th] | 2.73 (1.08), 646 | .157 (640) | 2.26 (0.89), 642 | .085 (636) |
| March 28[th] | 2.75 (1.05), 634 | .195 (627) | 2.34 (0.91), 634 | .089 (627) |
| April 11[th] | 2.56 (1.03), 547 | .118 (540) | 2.34 (0.95), 547 | .141 (540) |
| June 16[th] | 2.47 (0.97), 456 | .158 (452) | 2.17 (1.10), 456 | .183 (452) |

*Note*. Higher scores on this measure of political orientation correspond to more liberal attitudes.

behaviors were assessed at all five time points using the statement: "To minimize my chances of getting Coronavirus, I . . ." was followed by the items ". . .wash my hands more often.", ". . .avoid crowded spaces.", and ". . .put myself in quarantine." (-3 = *Strongly disagree*; +3 = *Strongly agree*). The items were specifically phrased to contextualize the behaviors as relevant to COVID-19 and were chosen because they covered the primary health-protective behaviors recommended by the WHO and the CDC at that time. Items were averaged to build a health-protective behaviors scale. The scale had satisfactory internal consistency (αs from .69 to .84 across time points). Descriptive statistics at each wave are presented in Table 2.

*Political orientation*. Prior research on COVID-19 suggests that single-item indicators of political orientation suffice to predict virus threat perceptions [12]. Political orientation was measured at Baseline with the item: "What is your political orientation?" (1 = *Extremely conservative*; 9 = *Extremely liberal*; M = 5.72, SD = 2.39).

## Results

**Political orientation and perceived health risk.** To examine whether political orientation was associated with perceived health risk, we calculated correlations between political orientation at Baseline and perceived health risk at all five time points. These correlations (see Table 2) show consistently across all time points that the more participants describe their political orientation at baseline as conservative, the lower they perceived their risk of infection. We also calculated partial correlations between the focal variables, controlling for gender, age, and education separately, as well as controlling for all three variables concurrently. The pattern of correlations between political orientation and perceived health risk was not altered after controlling for these variables.

**Political orientation and health-protective behaviors.** To examine whether political orientation was associated with health-protective behaviors, we calculated correlations between political orientation at Baseline and WHO-recommended health-protective behaviors at all five time points. The correlations depicted in Table 2 show a small but consistent pattern over time: the more participants described themselves as conservative, the less they enacted health-protective behaviors. In addition, partial correlations controlling for gender, age, and education separately, as well as for all three variables simultaneously, produced the same pattern of results.

**Mediation analyses.** To examine whether perceived infection risk mediated the relationship between political orientation and health-protective behaviors, we conducted five bootstrapping analyses (PROCESS macro, Model 4, 5,000 bootstrap samples [38]), one for each assessment wave. Note that political orientation was measured at Baseline, whereas perceived health risk and health-protective behaviors were measured at each time point. In support of

**Table 3. Tests of the mediational model in five time points: Study 1.**

| Date | Direct Effect: Baseline Political Orientation to WHO Virus Mitigation Behaviors | | | Indirect Effect: Baseline Political Orientation to WHO Virus Mitigation Behaviors through Perceived Risk | | |
|---|---|---|---|---|---|---|
| | *B* | *SE* | CI | *ab* | *SE* | CI |
| March 10th | .037 | .014 | .009, .064 | .004 | .002 | .001, .009 |
| March 20th | .028 | .015 | -.001, .057 | .003 | .003 | -.001, .010 |
| March 28th | .024 | .015 | -.006, .054 | .009 | .004 | .003, .018 |
| April 11th | .047 | .017 | .016, .081 | .006 | .003 | .001, .015 |
| June 16th | .072 | .020 | .031, .112 | .009 | .005 | .002, .020 |

*Note*. CI = 95% bootstrap confidence interval. The a and b pathways are presented in S1 Table.

our hypothesis, we found indirect effects of political orientation on health-protective behaviors via perceived health risk for four out of five assessment waves (see Table 3; additional path coefficients are presented in S1 Table).

Results suggest that in the U.S. context, political orientation at Baseline predicted health risk perceptions as well as health-protective behaviors across time. The finding that the association between these variables did not weaken during this three-months period is consistent with evidence that political orientation is stable over time, [39–41]. We also found evidence that health risk perceptions mediated the effects of political orientation on health-protective behaviors across time. However, a limitation of this study is that it is focused only on behaviors initially recommended by the WHO, whereas other behaviors–such as mask wearing and vaccination intentions, may have become more politicized during the course of the pandemic.

Moreover, American MTurk samples are not representative of Americans in general. MTurk workers tend to have lower average income, lower average ages and higher levels of education than the general population. MTurk samples are also more liberal than nationally representative samples [42, 43]. However, these factors would not be expected to change the associations between political orientation and compliance with health-protective behavior recommendations, or associations with the mediator variables that we examined.

Additionally, the present results were exclusive to the U.S. context, whereas a pandemic is a global phenomenon. Without comparing these patterns across countries, it is difficult to discern whether such patterns are due to differences in worldviews inherent to political ideologies, or if they are due to politicized influences that are unique to the United States. Finally, this study did not examine two other facets of health models: the perceived severity of COVID-19 as a mediator, or the use of face coverings or willingness to be vaccinated against COVID-19. To address these limitations, we report analyses from a second study in which participant recruitment extended beyond MTurk and beyond individuals currently living in the U.S.

## Study 2

The data we collected for Study 1 only allowed us to test the mediating role of perceived risk. However, health belief models also specify the perceived consequences of an infection and effectiveness of health-protective behaviors predict outcomes and both of these factors may have been politicized in the U.S. Additionally, in Study 1, we did not test whether the association between political orientation and health-protective behaviors was specific to the situation in the United States. We did, however, address these questions in Study 2. We also extended the health-protective behaviors we assessed to include wearing a face covering in public as it became more clearly recommended by the WHO (and the CDC). Although vaccines had not

yet been approved for use at the time of conducting this study, we anticipated (correctly) that vaccination would become a politically polarized topic, and thus also investigated vaccination intentions [44].

## Method

**Participants and procedure.** Participants from the U.S., as well as from 114 other countries were recruited for a longitudinal survey; S2 Table reports the most frequently represented countries at Baseline. Assessments began on March 19[th], 2020 and the current results are based on data collected up to July 13[th], 2020 from 62,909 individuals. The survey was distributed online through a combination of convenience sampling and snowball sampling. Members of the research team distributed the survey using social media campaigns, academic networks, and press releases. This convenience sample was supplemented with age and gender paid representative samples from 25 countries (collected only at baseline). On completion of the survey and debrief, a final screen invited respondents (both paid and unpaid) to distribute the survey to their networks and to participate in weekly (unpaid) follow-ups. To maximize data collection while minimizing participant strain, follow-up surveys with rotating questions were administered from March 19[th] to July 13[th], 2020. As new themes emerged in the discourse surrounding COVID-19, additional items were included in the survey. For instance, attitudes and behaviors pertaining to the wearing of masks/face coverings were added as the WHO amplified its support for their use.

Participants were eligible to enroll in the study by completing baseline at any point. Demographic characteristics of participants at Baseline are reported in Table 1. Most participants (75.43%) enrolled in the study between March 19[th] and April 18[th]; see S1 Fig for a histogram of date participants completed the Baseline survey. Following completion of the Baseline survey, participants received invitations to complete follow-up surveys at fixed time points (no follow-up surveys included participant payment). Some participants completed later follow-up waves but were not assessed in earlier follow-up waves because they entered the study only after those earlier follow-up waves had been administered. As a result of these design features, each wave contains both different subsets of the total sample of participants and differing time lag between baseline completion and follow-up survey completion, largely as a function of when participants enrolled in the study. In the U.S. sample, the timing of participants' completion of the baseline survey was not associated with political orientation ($r = .01$); within the non-U.S. sample, there was a small association of political orientation and enrollment in baseline study ($r = .11$).

Being a large-scale project covering a broad-range of psychological factors (for a full codebook of all questions included in the manuscript, see: https://osf.io/qhyue/), effect sizes for the research questions examined in this paper were not estimated a priori. All participants provided electronic consent in lieu of documenting signatures for consent. The study was approved by the Ethics Committees of the University of Groningen (PSY-1920-S-0390) and New York University Abu Dhabi (HRPP-2020-42).

**Measures.** *Political orientation.* We assessed political orientation using the image from the political compass (https://www.politicalcompass.org/analysis2). The official measure uses a lengthy text description to explain the graphic. For the purposes of the present study, we used the left to right continuum to capture conservatism without lengthy explanation. This measure was chosen for its adaptability across diverse political frameworks. Participants were specifically prompted to click on a position on the graphic that represents their political orientation from economically left (-200) to economically right (+200; $M_{US} = -16.04$, $SD = 80.68$; $M_{non-US} = -4.83$, $SD = 67.03$). In order to maintain consistency, we used the labels "conservative" and

"liberal" to refer to economic right and left orientations, respectively. As political orientation is recognized to be stable over time, we collected it only during the baseline survey [39–41].

*Perceived risk of infection.* As in Study 1, we assessed the perceived risk of infection using a single item about participants' perceived likelihood of becoming infected with coronavirus in the next few months (1 = *exceptionally unlikely*; 7 = *all but certain*). An additional response choice allowed participants to indicate if they had already become infected with the coronavirus. As the analyses focused on *perceptions* of risk, participants who selected this latter response were excluded from analyses. This measure of risk perception taps into the deliberative aspects of risk perceptions [45]. Although not purely objective, it assesses a threat-specific perception of likelihood. We assessed perceived risk in the baseline survey and in nine follow-up surveys.

*Perceived severity of infection.* To capture subjective perceptions of risk, we asked participants to indicate how subjectively disturbing it would be for them if they were infected with Coronavirus (1 = *not disturbing at all*; 5 = *extremely disturbing*). This measure represents an experiential health risk perception that combines broad affective responses to the trigger (e.g., stigma about becoming infected, fear of the side effects of the disease) and deliberative aspects of risk (e.g., awareness of increased risk with age or employment status [45]). Perceived severity was assessed only in the baseline survey.

*Perceived effectiveness of health-protective behaviors.* Beliefs about health-protective behaviors being effective in protecting against the risk of infection were assessed using two separate measures about social distancing (at three time points) and wearing a mask (at four time points). Participants reported their beliefs about the effectiveness of social distancing by agreeing with the statement 'In the absence of effective medical treatment or vaccines, social distancing measures are the most effective means of controlling the pandemic'. Participants reported beliefs about the effectiveness of wearing a mask or face covering for preventing infection of COVID-19 by indicating their agreement with the statement 'I believe that wearing a mask protects myself.' Both efficacy belief items used the same scale (-2 = *strongly disagree*; +2 = *strongly agree*). Perceptions of the effectiveness of social distancing were measured in three follow-up surveys; perceptions of the effectiveness of wearing a face covering were measured in four follow-up surveys.

*Health-protective behaviors.* We assessed three health-protective behaviors. Correlations between each health behavior in the follow-up surveys were small to medium, (see S3 Table).

*WHO virus mitigation behaviors.* As in Study 1, we assessed engagement in the three health-protective behaviors recommended by the WHO (i.e., hand washing, avoiding crowds, and self-isolating). We reused the items and scaling from Study 1. Our primary concern was utilizing a set of items to capture adherence to behaviors that were uniformly recommended for virus mitigation. The items demonstrated acceptable internal consistency (αs = .62-.74). WHO Virus mitigation behaviors were measured at baseline and in three follow-up waves.

*Willingness to be vaccinated.* Participants reported their willingness to be vaccinated in three follow-up waves by responding to the question 'How likely are you to get vaccinated against coronavirus once a vaccine becomes available?' on a five point scale (-2 = *extremely unlikely*; +2 = *extremely likely*). The item was adapted from prior flu vaccine research using a single item measure to capture vaccine intentions [46]. Note that the final assessment of vaccine intentions was conducted in July 2020, well before any vaccine had been approved for use.

*Wearing a mask.* Although wearing a face mask is now considered a health-protective behavior, it was not initially recommended by the WHO and was, therefore, neither included at Baseline, nor at any time point in Study 1. As it became evident that mask wearing would be a critical health-protective behavior in response to the COVID-19 pandemic, we added a

measure of it to our longitudinal survey. At four time points (Waves 6, 8, 10, and 12), partici-pants were asked about their frequency of wearing a mask/face covering in public. Participants responded to the statement 'In the past week, I have covered my face in public places,' using a five point scale (1 = [almost] never; 5 = [almost] always).

## Results

**Data analytic plan.** Given the large sample size differences between participants who par-ticipated in our baseline and follow-up surveys, we analyzed the baseline and follow-up data separately. To simplify presentation of results and account for measuring different variables at different times, we averaged responses to the same variable across the follow-up waves. These averages estimate participants' relatively enduring standing on each variable. The primary analyses for the follow-up waves were conducted using these averages. Means and sample sizes for each variable across wave of data collection are presented in Table 4. As shown in Table 4, each participant's average score reflects between one and four responses, depending on the number of waves the measure was assessed (e.g., WHO Virus Mitigation behaviors were assessed in three follow-up waves) and the number of waves the participant completed. We report results for each follow-up wave separately in the supplementary materials.

We first evaluated the zero-order correlations between political orientation and each out-come (i.e., perceived risk, health-protective behaviors) within and across locations (U.S. vs non U.S.). To compare correlations across locations, we used a general linear model with loca-tion (U.S. = 0; non-U.S. = 1) as a categorical between-subjects factor and political orientation as a continuous between-subjects factor. A test of the interaction between location and political orientation evaluated whether associations between political orientation and outcomes were different for participants living inside versus outside of the United States. These parsimonious models allow for easy interpretation of effects and effect size.

We also conducted robustness checks to confirm that the interaction between location and political orientation persisted after several considerations. First, political orientation was weakly associated with age ($r$ = .04), education ($r$ = -.09), and gender ($r$ = -.04) at Baseline, and these factors might be expected to account for some of the shared variance between political orientation and health beliefs and behaviors. Additionally, we observed that date of Baseline survey completion was related to most outcomes (see S4 Table). Finally, participants in our study were not entirely independent of each other—people residing within different countries were exposed to different messaging, norms, and support factors related to COVID-19. Thus, we conducted robustness checks using multilevel modeling in which participants were nested in countries (with intercepts modeled as randomly varying across countries), while controlling for age, education, gender, and baseline survey date. Cumulatively, these robustness checks allowed us to account for interdependence of data and the alternative explanation that percep-tions of risk might be due to demographic or methodological (i.e., differential enrollment across time) factors. All observed interactions between location (U.S. vs non-U.S.) and political orientation remained significant after these robustness checks (see S4 Table).

**Political orientation and perceived health risk.** As Table 5 shows, in the U.S., political orientation was associated with perceived risk of infection such that more conservative indi-viduals reported a lower likelihood of becoming infected. We tested whether correlations in the U.S. and non-U.S. sampled by evaluating the interaction term between political orientation and location; these tests are reported in Table 5 with both an $F$ value representing the interac-tion and an effect size representing the proportion of variance in the outcome explained by the interaction between the variables. The correlations between political orientation and perceived risk were stronger in the U.S. than in the non-U.S. sample. The observed interactions between

**Table 4. Dates, participants, and descriptive statistics of variables used in analyses, Study 2.**

| | BL | W1 | W2 | W3 | W4 | W5 | W6 | W7 | W8 | W9 | W10 | W11 | W12 | Ave |
|---|---|---|---|---|---|---|---|---|---|---|---|---|---|---|
| **Political Orientation** | | | | | | | | | | | | | | |
| U.S. | 16.06 (80.68) | -36.91 (84.42) | -24.44 (84.29) | -25.93 (84.93) | -24.79 (83.20) | -31.35 (83.34) | -30.15 (82.24) | -25.54 (83.05) | -31.79 (81.89) | -32.10 (82.70) | -35.50 (80.21) | -36.84 (80.84) | -36.23 (80.15) | |
| | 10923 | 540 | 2672 | 1856 | 1356 | 1031 | 883 | 601 | 803 | 743 | 527 | 769 | 689 | |
| Non-U.S. | -4.83 (67.03) | -42.42 (72.07) | -35.69 (69.89) | -35.37 (71.00) | -17.17 (73.29) | -16.99 (72.91) | -18.33 (72.77) | -18.44 (73.05) | -20.07 (73.03) | -18.48 (73.21) | -21.00 (71.99) | -19.57 (72.01) | -20.33 (72.38) | |
| | 51986 | 981 | 3514 | 3621 | 6588 | 6251 | 5014 | 4651 | 4282 | 4052 | 3391 | 4128 | 3596 | |
| **Perceived Risk of COVID-19** | | | | | | | | | | | | | | |
| U.S. | 3.78 (1.38) | 3.98 (1.29) | 3.76 (1.32) | 3.67 (1.32) | | 3.66 (1.40) | | 3.67 (.32) | | 3.64 (1.32) | | 3.69 (1.34) | 3.84 (1.25) | 3.73 (1.27) |
| | 10912 | 540 | 2672 | 1856 | | 1031 | | 601 | | 743 | | 769 | 689 | 4166 |
| Non-U.S. | 3.48 (1.40) | 4.12 (1.41) | 3.90 (1.30) | 3.85 (1.35) | | 3.61 (1.36) | | 3.59 (1.35) | | 3.48 (1.34) | | 3.51 (1.34) | 3.67 (1.29) | 3.61 (1.27) |
| | 51750 | 981 | 3514 | 3621 | | 6251 | | 4651 | | 4052 | | 4128 | 3596 | 12901 |
| **Perceived Severity of COVID-19** | | | | | | | | | | | | | | |
| U.S. | 4.04 (1.14) | | | | | | | | | | | | | |
| | 10914 | | | | | | | | | | | | | |
| Non-U.S. | 3.87 (1.28) | | | | | | | | | | | | | |
| | 51684 | | | | | | | | | | | | | |
| **WHO Virus Mitigation Behaviors** | | | | | | | | | | | | | | |
| U.S. | 2.22 (0.95) | | | | 2.09 (1.13) | | | | | | | 1.69 (1.31) | 1.79 (1.29) | 1.94 (1.15) |
| | 10917 | | | | 1357 | | | | | | | 769 | 689 | 1811 |
| Non-U.S. | 2.20 (0.99) | | | | 2.10 (1.07) | | | | | | | 1.28 (1.38) | 1.13 (1.40) | 1.71 (1.20) |
| | 51805 | | | | 6590 | | | | | | | 4127 | 3600 | 8621 |
| | BL | W1 | W2 | W3 | W4 | W5 | W6 | W7 | W8 | W9 | W10 | W11 | W12 | Ave |
| **Perceived Efficacy of Social Distancing** | | | | | | | | | | | | | | |
| U.S. | | 1.50 (0.78) | 1.53 (0.77) | 1.50 (0.83) | | | | | | | | | | 1.50 (0.75) |
| | | 2672 | 1856 | 1357 | | | | | | | | | | 3576 |
| Non-U.S. | | 1.36 (0.84) | 1.34 (0.83) | 1.39 (0.81) | | | | | | | | | | 1.36 (0.80) |
| | | 3513 | 3620 | 6588 | | | | | | | | | | 8987 |
| **Vaccine Intentions** | | | | | | | | | | | | | | |
| U.S. | | | | | 1.16 (1.19) | | | | | | | 1.13 (1.23) | | 1.14 (1.17) |
| | | | | | 1357 | | | | | | | 769 | | 1811 |
| Non-U.S. | | | | | 1.39 (0.81) | | | | | | | 0.82 (1.24) | | 0.86 (1.18) |
| | | | | | 6588 | | | | | | | 4098 | | 8519 |
| **Efficacy of Wearing a Face Covering** | | | | | | | | | | | | | | |
| U.S. | | | | | | 0.67 (1.28) | | 0.58 (1.31) | | 0.64 (1.27) | | | 1.21 (1.15) | 0.75 (1.19) |
| | | | | | | 960 | | 834 | | 549 | | | 689 | 1489 |
| Non-U.S. | | | | | | 0.51 (1.31) | | 0.33 (1.34) | | 0.52 (1.31) | | | 0.83 (1.23) | 0.58 (1.23) |
| | | | | | | 5553 | | 4484 | | 3572 | | | 3576 | 7659 |
| **Wearing a Face Covering** | | | | | | | | | | | | | | |

*(Continued)*

**Table 4.** (*Continued*)

| | BL | W1 | W2 | W3 | W4 | W5 | W6 | W7 | W8 | W9 | W10 | W11 | W12 | Ave |
|---|---|---|---|---|---|---|---|---|---|---|---|---|---|---|
| **U.S.** | | | | | | | 4.22 (1.31) | | 4.32 (1.23) | | 4.38 (1.18) | | 4.70 (0.83) | 4.38 (1.11) |
| | | | | | | | 883 | | 803 | | 527 | | 646 | 1441 |
| **Non-U.S.** | | | | | | | 3.47 (1.66) | | 3.59 (1.60) | | 3.61 (1.60) | | 3.72 (1.49) | 3.64 (1.50) |
| | | | | | | | 5014 | | 4484 | | 3391 | | 3310 | 7349 |

Notes. Political orientation was assessed *only* at baseline. Presented numbers reflect the political orientation (reported at baseline) of participants who completed each wave. BL = baseline, W = wave, Ave = average of construct across wave.

location and political orientation remained similar during our multilevel model robustness checks that nested participants within each location and controlled for date of survey completion, age, education, and gender at all time points.

In our cross-sectional Baseline questionnaire, political orientation was also negatively associated with expected severity of infection such that more conservative individuals expected a COVID-19 infection to be less severe if they were to contract it. Here, the association of political orientation and perceived severity reversed direction for participants living outside the U.S.

**Table 5. Correlations between baseline political orientation and perceived risk, perceived effectiveness, and health-protective behaviors, Study 2.**

| | U.S. *r* | Non-U.S. *r* | U.S. vs. Non-U.S. Correlation Comparison |
|---|---|---|---|
| | | | *F* |
| | N | N | $\eta^2$ (90%CI) |
| **Baseline** | | | |
| **Perceived Risk** | -.13*** | -.08*** | *F* = 5.87* |
| | 10912 | 51570 | $\eta^2 < .001$ (< .001, < .001) |
| **Severity of Contracting the Virus:** | -.08*** | .03*** | *F* = 87.65*** |
| | 10914 | 51684 | $\eta^2 = .001$ (.001, .002) |
| **WHO Virus Mitigation Behaviors** | -.13*** | -.03*** | *F* = 76.38*** |
| | 11030 | 52072 | $\eta^2 = .001$ (.001, .002) |
| **Follow-up Waves** | | | |
| **Perceived Risk** | -.19*** | -.08*** | *F* = 26.64*** |
| | 4166 | 12901 | $\eta^2 = .002$ (.001, .003) |
| **Effectiveness of Social Distancing** | -.22*** | -.02 | *F* = 82.49*** |
| | 3576 | 8987 | $\eta^2 = .006$ (.004, .009) |
| **Effectiveness of Wearing a Mask** | -.17*** | .08*** | *F* = 73.38*** |
| | 1489 | 7659 | $\eta^2 = .008$ (.005, .011) |
| **WHO Virus Mitigation Behaviors** | -.23*** | .02 | *F* = 82.64*** |
| | 1811 | 8621 | $\eta^2 = .008$ (.005, .011) |
| **Wearing a Mask** | -.28*** | .04** | *F* = 76.28*** |
| | 1441 | 7349 | $\eta^2 = .008$ (.005, .012) |
| **Willingness to be Vaccinated** | -.32*** | -.07*** | *F* = 84.45*** |
| | 1811 | 8519 | $\eta^2 = .008$ (.005, .011) |

*$p < .05$

**$p < .01$

***$p < .001$.

*Note*. This table reports the results of averaged responses across the Follow-Up waves. Results across each time point were consistent and can be seen in S5 Table.

**Political orientation and perceived effectiveness of health-protective behaviors.** Political orientation was also associated with the perceived effectiveness of health-protective behaviors (i.e., social distancing, wearing a face covering; see Table 4) such that more conservative individuals perceived these behaviors as less useful. The interactions with location indicate that these effects were stronger for participants in the U.S. relative to non-U.S. participants, and these interactions persisted after our robustness checks.

**Political orientation and health-protective behaviors.** As Table 5 shows, political orientation was associated with the WHO recommended health-protective behaviors, willingness to be vaccinated, and wearing a face covering, such that more conservative individuals engaged in less health-protective behaviors. These effects were larger among U.S. participants versus non-U.S. participants at every time point, and the interactions between political orientation and location held during our robustness checks.

**Mediation analyses.** As in Study 1, we used the PROCESS macro (seed = 31216 [38]) to examine whether the relationship between political orientation and health-protective behaviors was mediated by perceived health risk, perceived severity of contracting the virus, or perceived efficacy of health-protective behaviors (Model 4), as well as whether indirect effects were moderated by location (Model 7). We conducted two sets of analyses—one for measures assessed only at baseline and one for measures assessed at follow-up. Note that due to its identified stability over time [39–41], political orientation was only measured at Baseline. For all mediational analyses, we standardized the political orientation variable. Because all correlations and interactions between location and political orientation remained after our robustness checks, tests of indirect effects neither included covariates nor nested participants into location.

Across all health-protective behaviors and mediators, we observed consistent patterns of a) mediation of health-protective behaviors by perceived risk, perceived consequences, and effectiveness of relevant health-protective behaviors and b) stronger indirect effects for U.S. participants relative to non-U.S. participants (see Table 6). Notably, the association between political orientation and perceived effectiveness of wearing a face covering were negative for participants living in the United States, but positive for participants living outside the U.S., consistent with our hypothesis regarding the unique effects of political orientation in the U.S., and reflecting the potentially unique discourse surrounding masks in the U.S.

**Ancillary analyses.** In the analyses described above, we compared associations across US and non-US participants living in 114 other countries. To examine the possibility that our findings could be an artifact of aggregating the data across these 114 countries, we also explored these effects in a (sub-)sample of individual countries. We focused our attention to comparison countries in which we recruited the most participants into the Baseline survey (Spain [*n* = 3156], Romania [*n* = 2696], Netherlands [*n* = 2992]), Indonesia [*n* = 2407], Greece [*n* = 2870], and Republic of Serbia [*n* = 2118]). Additionally, we evaluated responses in Canada (*n* = 1531) because residents of Canada might be expected to be exposed to political messaging from the U.S. to a greater degree than other individuals due to the proximity and shared border of the countries, and thus represent a conservative examination of the unique effects of political orientation in the U.S. Decisions about which countries to include were made prior to examining direction or size of associations within the comparison countries. Both total and partial (i.e., controlling for age, gender, education, and date of baseline survey completion) associations between political orientation and health beliefs and behaviors were larger in the U.S. than in each of these other comparison countries, with the exception that several associations were similar across the U.S., Republic of Serbia, and Canada. S8 Table reports the associations within the U.S. and these seven comparison countries.

**Table 6. Perceived risk and severity mediates the relationship between baseline political orientation and health-protective behaviors, Study 2.**

| | IV to Mediator | Mediator to DV | Direct Effect | Indirect Effect |
|---|---|---|---|---|
| | *b* (CI) | *b* (CI) | *b* (CI) | *b* (CI) |
| **Baseline** | | | | |
| **Baseline Political Orientation–Perceived Risk—WHO Virus Mitigation Behaviors** | | | | |
| | | Index of Moderated Mediation: .0014 (.0002, .0025) | | |
| US | -.153 (-.175, -.131) | .023 (.010, .036) | -.110 (-.122, -.010) | -.0065 (-.0079, -.0052) |
| Non-US | -.121 (-.134, -.109) | .046 (.040, .052) | -.023 (-.032, -.014) | -.0052 (-.0061, -.0043) |
| **Baseline Political Orientation–Perceived Severity—WHO Virus Mitigation Behaviors** | | | | |
| | | Index of Moderated Mediation: .0201 (.0162, .0241) | | |
| US | -.077 (-.096, -.059) | .224 (.209, .239) | -.093 (-.107, -.078) | -.0140 (-.0174, -.0106) |
| Non-US | .034 (.022, .045) | .173 (.167, .179) | -.034 (-.043, -.025) | .0061 (.0041, .0082) |
| **Follow-Up Waves** | | | | |
| **Baseline Political Orientation–Perceived Risk—WHO Virus Mitigation Behaviors** | | | | |
| | | Index of Moderated Mediation: .0140 (.0076, .0206) | | |
| US | -.223 (-.269, -.176) | .098 (.055, .141) | -.199 (-.244, -.155) | -.0245 (-.0319, -.0176) |
| Non-US | -.096 (-.120, -.081) | .106 (.085, .126) | .019 (-.005, .044) | -.0105 (-.0141, -.0073) |
| **Baseline Political Orientation–Perceived Risk—Willingness to be Vaccinated** | | | | |
| | | Index of Moderated Mediation: .0204 (.0115, .0301) | | |
| US | -.223 (-.269, -.176) | .196 (.154, .238) | -.275 (-.319, -.232) | -.0367 (-.0466, -.0280) |
| Non-US | -.099 (-.123, -.074) | .151 (.131, .172) | .-.064 (-.088, -.040) | -.0162 (-.0210, -.0118) |
| **Baseline Political Orientation–Perceived Risk—Wearing a Mask** | | | | |
| | | Index of Moderated Mediation: .0226 (0.0124, .0341) | | |
| US | -.233 (-.288, -.178) | .163 (.115, .210) | -.235 (-.285, -.185) | -.0350 (-.0465, -.0248) |
| Non-US | -.083 (-.111, -.055) | .131 (.101, .160) | .051 (.016, .085) | -.0124 (-.0177, -.0080) |
| **Baseline Political Orientation–Perceived Efficacy of Wearing a Mask—Wearing a Mask** | | | | |
| | | Index of Moderated Mediation: .1514 (.1148, .1894) | | |
| US | -.172 (-.224, -.121) | .372 (.329, .415) | -.199 (-.242, -.155) | -.0986 (-.1315, -.0662 |
| Non-US | .092 (.065, .119) | .590 (.566, .615) | -.003 (-.031, .026) | .0527 (.0378, .0686) |
| **Baseline Political Orientation–Perceived Efficacy of Social Distancing—WHO Virus Mitigation Behaviors** | | | | |
| | | Index of Moderated Mediation: .0691 (.0499, .0894) | | |
| US | -.158 (-.188, -.128) | .690 (.622, .758) | -.098 (-.140, -.055) | -.0797 (-.0981, -.0623) |
| Non-US | -.021 (-.038, -.004) | .455 (.422, .488) | .041 (.162, .065) | -.0106 (-.0197, -.0012) |

*Notes*. Results across each time point were consistent, for analyses within time point, see S6 and S7 Tables. Political orientation was standardized prior to analysis. We analyzed the indirect pathway between political orientation and WHO virus mitigation behaviors through perceived efficacy of social distancing because two of the three items included in that scale relate to keeping distance from others.

## General discussion

Countrywide lockdowns and social distancing measures have severe economic and social consequences. Because compliance with behavioral recommendations involves personal costs, people need to be persuaded that it is in their own best interest to adopt these behaviors. According to theories of health behavior, this can be achieved by convincing people that there is a high risk of getting infected, that this has serious, often fatal consequences, and that recommended health-protective behaviors will in fact be effective in reducing the infection risk [3–6, 9, 10].

The U.S. is one of few countries where leading conservative government figures as well as an influential conservative-leaning news network questioned both the seriousness of the pandemic and the effectiveness of some of the recommended health-protective behaviors. As

empirical studies have already demonstrated, U.S. conservatives engaged in less health-protective behaviors related to COVID-19 than did U.S. liberals. Recent work suggests these patterns have extended over time, even during periods of increased disease threat [47]. Our work replicates these patterns and suggests that perceptions of risk and effectiveness of health behaviors partially explain the effects of political orientation on enactment of these behaviors. Effects of political orientation were stronger (and sometimes in the opposite direction) in the United States than they were globally, which provides novel evidence suggesting these political differences are explained by politicized forces within the United States rather than differences in beliefs fundamental to political ideologies.

We were fortunate to be able to capture perceptions of health risk and health-protective behaviors as the pandemic was beginning to unfold internationally. By following up with participants over time, we were able to assess associations between political ideology, risk perceptions, and health-protective behaviors, even as the context of the pandemic changed. In Study 2, we could observe beliefs about risk and the emerging health-protective behaviors of wearing a face covering and intentions to get vaccinated. Across both studies, we observed strong consistency in the size of effects over time, suggesting that differences due to political affiliation emerged due to early politicization of health-protective behaviors. Indeed, during March and April 2020, these differences were already prominent between conservative and liberal political leaders.

These patterns are consistent with our prediction that the deleterious effects of political orientation on health-protective behaviors are specific to the U.S. and to the conservative leadership during the early stages of the pandemic. Indeed, outside of the U.S., conservatives were more likely than liberals to believe masks would provide personal protection (and were consequently more likely to report wearing a face covering). Moreover, these patterns are different than what might be expected based on evidence that conservatives are more sensitive to threats (especially physical threats) than liberals [48, 49] and that conservatives in the U.S. (relative to liberals in the U.S.) expressed more concern about a pandemic happening under other (Democratic) political leadership [24]. Thus, although we did not empirically assess attention to or agreement with conservative leadership and news sources, the patterns we observe differ from what might be expected based on past research on conservative responses to virus threats, suggesting a U.S.-specific and COVID-specific influence. Although our studies did not directly examine political communication, their findings highlight mechanisms by which political communication may become life-threatening—when it alters the perceptions of risk of health-threatening circumstances and the efficacy of mitigation behavior.

Another strength of our studies is the size of our samples and their repeated measures over time. Because we captured health behaviors and perceptions at many points in a changing pandemic, it is unlikely that associations between baseline orientation and outcomes were driven by one specific contextual factor. Admittedly, the effect sizes representing the association between political orientation and compliance with health-protective behavior recommendations observed in our samples are small. Our large samples allowed us to identify these small effect sizes precisely, as noted by narrow confidence intervals. Moreover, weak effects on an individual level can still have a powerful impact at the population level. For example, even though smokers run a much greater risk of lung cancer than non-smokers, the 10-year absolute risk of lung cancer for a 35-year old man who is a heavy smoker is only about 0.9% [50]. And yet, these small effects have a great impact at the population level. In a group of 1 million heavy smokers aged 35, for instance, nearly 10,000 will die prematurely before the age of 45 due to smoking [50].

As all research, our research has some limitations. Because our data are correlational, we cannot draw causal conclusions. What we can show, however, is that the pattern of our data is

consistent with such a causal interpretation. And such support is evidenced by the consistent mediation of health-protective behaviors by perceived risk of getting infected, perceived severity of the consequences of such an infection, and perceived efficacy of relevant health-protective behaviors in preventing such negative outcomes. Finally, the difference in the magnitude of the indirect effects between the U.S. and the non-U.S. data suggest that these effects are specific to the situation in the U.S. Table 5, which compares effects across U.S. and non-US participants, illustrates the uniqueness of the U.S. effects. We also found parallel patterns for the perceived effectiveness of social distancing. Political orientation predicted willingness to observe social distancing, and this association was mediated by perceived effectiveness, with the mediation effect again being moderated by location.

Another weakness is that our measures of political orientation, particularly in Study 2, are not optimal. Whereas Republican-leaning Americans are likely to rate themselves as more conservative than Democrat-leaning Americans, this association is less clear for the economic dimension of the political compass used in Study 2. The political compass measure was chosen in order to make the political orientation data comparable across countries. However, the fact that the correlations are of a similar magnitude in both studies suggests that the right to left dimension was similar to the conservative to liberal continuum. Most likely, Republican-leaning conservatives would identify as right-leaning relative to Democratic-leaning liberals. However, not all conservatives are Republicans, Trump supporters, or viewers of Fox News, and these characteristics would more directly index exposure to messages that have downplayed COVID-19, as well as susceptibility to influence by such messages. In a future study, we would also assess participants on the dimension of conservatism to liberalism to ascertain the correlation of this dimension with the political compass. A further potential weakness is the fact that we did not measure all variables concurrently during all waves. These decisions were made to conserve space and reduce participant burden but limit causal interpretations. Finally, the fact that—as a function of when participants enrolled in the study—each wave in our sample contained both different subsets of the total sample of participants and differing time lag between baseline completion and follow-up survey completion, complicated our analysis. In a future study, we would definitely avoid this problem, even at the cost of being able to enroll fewer participants.

Our studies illustrate both the applicability of social- and health-psychological theories to address a real-world issue and the use of a real-world problem to test psychological theories. The starting point of our analyses is the political situation in the U.S., where the former president as well as leading conservative politicians consistently downplayed the severity of the COVID-19 pandemic and belittled the effectiveness of scientific recommendations regarding health-protective behaviors. Like other researchers and opinion surveys before us, we showed that political orientation is associated with compliance with recommended health-protective behaviors. We expanded on this research by testing and supporting the theory-based prediction that this association was mediated by risk perception, perceived severity of the infection as well the perceived effectiveness of the recommended health-protective behaviors.

Individuals who fail to comply with health-protective behavior recommendations increase their chances of contracting COVID-19, dying or suffering long-term effects from the disease, and spreading it to others. Our studies suggest that politicized messages from leaders and media outlets that downplay risks might be linked to increased spread of COVID-19. Indeed, U.S. counties that voted for Donald Trump over Hillary Clinton in 2016 have not only observed less social distancing, but this failure to observe social distancing was associated with subsequently higher COVID-19 infections and fatalities [16].

## Supporting information

**S1 Table Unstandardized coefficients for the paths from baseline political orientation to perceived health risk (a) and from perceived risk to health-protective behaviors (b) at five time points, Study 1.**
(DOCX)

**S2 Table. Number of participants from the most frequently represented countries, Study 2.**
(DOCX)

**S3 Table. Correlations between health-protective behaviors at follow-up, Study 2.**
(DOCX)

**S4 Table. Results of robustness checks in which participants were nested into country, and we included date of baseline survey completion, age, gender, and education as covariates.**
(DOCX)

**S5 Table. Correlations between political orientation and perceived risk, perceived efficacy, and health-protective behaviors.**
(DOCX)

**S6 Table. Perceived risk and severity mediates relationship between political orientation and health-protective behaviors separated by wave, Study 2.**
(DOCX)

**S7 Table. Perceived risk and efficacy mediates effects of political orientation on wearing a mask by wave, Study 2.**
(DOCX)

**S8 Table. Correlations between political orientation and other variables across specific comparison countries.**
(DOCX)

**S1 Fig. Distribution of days after March 19 (beginning of survey) in which participants completed the baseline.**
(TIF)

## Author Contributions

**Conceptualization:** Wolfgang Stroebe, Michelle R. vanDellen, Georgios Abakoumkin, Edward P. Lemay, Jr., William M. Schiavone, Maximilian Agostini, Jocelyn J. Bélanger, Ben Gützkow, Jannis Kreienkamp, Anne Margit Reitsema, Bertus F. Jeronimus, Arie W. Kruglanksi, Maja Kutlaca, Jolien Anne van Breen, Caspar J. Van Lissa, N. Pontus Leander.

**Data curation:** Allan B. I. Bernardo, Nicholas R. Buttrick, Ali Hamaidia, N. Pontus Leander.

**Formal analysis:** Michelle R. vanDellen, Georgios Abakoumkin.

**Funding acquisition:** Jocelyn J. Bélanger, Manuel Moyano, N. Pontus Leander.

**Investigation:** Wolfgang Stroebe, Maximilian Agostini, N. Pontus Leander.

**Methodology:** Wolfgang Stroebe, Michelle R. vanDellen, Maximilian Agostini, Jocelyn J. Bélanger, Ben Gützkow, Jannis Kreienkamp, Anne Margit Reitsema, Jamilah Hanum Abdul Khaiyom, Vjolica Ahmedi, Handan Akkas, Carlos A. Almenara, Mohsin Atta, Sabahat Cigdem Bagci, Sima Basel, Edona Berisha Kida, Allan B. I. Bernardo, Phatthanakit

Chobthamkit, Hoon-Seok Choi, Mioara Cristea, Sára Csaba, Kaja Damnjanović, Ivan Danyliuk, Arobindu Dash, Daniela Di Santo, Karen M. Douglas, Violeta Enea, Daiane Gracieli Faller, Gavan Fitzsimons, Alexandra Gheorghiu, Ángel Gómez, Ali Hamaidia, Qing Han, Mai Helmy, Joevarian Hudiyana, Bertus F. Jeronimus, Ding-Yu Jiang, Veljko Jovanović, Željka Kamenov, Anna Kende, Shian-Ling Keng, Tra Thi Thanh Kieu, Yasin Koc, Kamila Kovyazina, Inna Kozytska, Joshua Krause, Arie W. Kruglanksi, Anton Kurapov, Maja Kutlaca, Nóra Anna Lantos, Cokorda Bagus Jaya Lemsmana, Winnifred R. Louis, Adrian Lueders, Najma Iqbal Malik, Anton Martinez, Kira O. McCabe, Jasmina Mehulić, Mirra Noor Milla, Idris Mohammed, Erica Molinario, Manuel Moyano, Hayat Muhammad, Silvana Mula, Hamdi Muluk, Solomiia Myroniuk, Reza Najafi, Claudia F. Nisa, Boglárka Nyúl, Paul A. O'Keefe, Jose Javier Olivas Osuna, Evgeny N. Osin, Joonha Park, Gennaro Pica, Antonio Pierro, Jonas Rees, Elena Resta, Marika Rullo, Michelle K. Ryan, Adil Samekin, Pekka Santtila, Edyta Sasin, Birga M. Schumpe, Heyla A. Selim, Michael Vicente Stanton, Samiah Sultana, Robbie M. Sutton, Eleftheria Tseliou, Akira Utsugi, Jolien Anne van Breen, Caspar J. Van Lissa, Kees Van Veen, Alexandra Vázquez, Robin Wollast, Victoria Wai-Lan Yeung, Somayeh Zand, Iris Lav Žeželj, Bang Zheng, Andreas Zick, Claudia Zúñiga, N. Pontus Leander.

**Project administration:** Wolfgang Stroebe, Michelle R. vanDellen, Edward P. Lemay, Jr., Maximilian Agostini, Jocelyn J. Bélanger, Ben Gützkow, Jannis Kreienkamp, Anne Margit Reitsema, Vjolica Ahmedi, Handan Akkas, Carlos A. Almenara, Mohsin Atta, Sabahat Cigdem Bagci, Sima Basel, Edona Berisha Kida, Nicholas R. Buttrick, Phatthanakit Chobthamkit, Hoon-Seok Choi, Mioara Cristea, Sára Csaba, Kaja Damnjanović, Ivan Danyliuk, Arobindu Dash, Daniela Di Santo, Karen M. Douglas, Violeta Enea, Daiane Gracieli Faller, Gavan Fitzsimons, Alexandra Gheorghiu, Ángel Gómez, Qing Han, Mai Helmy, Joevarian Hudiyana, Bertus F. Jeronimus, Ding-Yu Jiang, Veljko Jovanović, Željka Kamenov, Anna Kende, Shian-Ling Keng, Tra Thi Thanh Kieu, Yasin Koc, Kamila Kovyazina, Inna Kozytska, Joshua Krause, Anton Kurapov, Maja Kutlaca, Nóra Anna Lantos, Cokorda Bagus Jaya Lemsmana, Winnifred R. Louis, Adrian Lueders, Najma Iqbal Malik, Anton Martinez, Kira O. McCabe, Jasmina Mehulić, Mirra Noor Milla, Idris Mohammed, Erica Molinario, Manuel Moyano, Hayat Muhammad, Silvana Mula, Hamdi Muluk, Solomiia Myroniuk, Reza Najafi, Claudia F. Nisa, Boglárka Nyúl, Paul A. O'Keefe, Jose Javier Olivas Osuna, Evgeny N. Osin, Joonha Park, Gennaro Pica, Antonio Pierro, Jonas Rees, Elena Resta, Marika Rullo, Michelle K. Ryan, Adil Samekin, Pekka Santtila, Edyta Sasin, Birga M. Schumpe, Heyla A. Selim, Michael Vicente Stanton, Samiah Sultana, Robbie M. Sutton, Eleftheria Tseliou, Akira Utsugi, Jolien Anne van Breen, Caspar J. Van Lissa, Kees Van Veen, Alexandra Vázquez, Robin Wollast, Victoria Wai-Lan Yeung, Somayeh Zand, Iris Lav Žeželj, Bang Zheng, Andreas Zick, Claudia Zúñiga, N. Pontus Leander.

**Resources:** William M. Schiavone, Maximilian Agostini, Kira O. McCabe, Solomiia Myroniuk, Samiah Sultana, Caspar J. Van Lissa, N. Pontus Leander.

**Supervision:** Wolfgang Stroebe, Jocelyn J. Bélanger, Arie W. Kruglanksi, Samiah Sultana, N. Pontus Leander.

**Visualization:** Michelle R. vanDellen, Georgios Abakoumkin.

**Writing – original draft:** Wolfgang Stroebe, Michelle R. vanDellen, Georgios Abakoumkin, Edward P. Lemay, Jr., N. Pontus Leander.

**Writing – review & editing:** Wolfgang Stroebe, Michelle R. vanDellen, Georgios Abakoumkin, Edward P. Lemay, Jr., William M. Schiavone, Maximilian Agostini, Jocelyn J. Bélanger,

Ben Gützkow, Jannis Kreienkamp, Anne Margit Reitsema, Sima Basel, Allan B. I. Bernardo, Karen M. Douglas, Ali Hamaidia, Bertus F. Jeronimus, Shian-Ling Keng, Maja Kutlaca, Winnifred R. Louis, Kira O. McCabe, Manuel Moyano, Solomiia Myroniuk, Claudia F. Nisa, Samiah Sultana, Robbie M. Sutton, Claudia Zúñiga, N. Pontus Leander.

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
