## [Decision Letter · Decision Letter 0]

5 Jun 2021

PONE-D-21-14147

Politicization of COVID-19 Health-Protective Behaviors in the United States: Longitudinal and Cross-National Evidence

PLOS ONE

Dear Dr. vanDellen,

Thank you for submitting your manuscript to PLOS ONE. After careful consideration, we feel that it has merit but does not fully meet PLOS ONE’s publication criteria as it currently stands. Therefore, we invite you to submit a revised version of the manuscript that addresses the points raised during the review process.

We look forward to receiving your revised manuscript.

Kind regards,

Amitava Mukherjee, ME, Ph.D.

Academic Editor

PLOS ONE

Journal Requirements:

2a) If there are ethical or legal restrictions on sharing a de-identified data set, please explain them in detail (e.g., data contain potentially sensitive information, data are owned by a third-party organization, etc.) and who has imposed them (e.g., an ethics committee). Please also provide contact information for a data access committee, ethics committee, or other institutional body to which data requests may be sent.

2b) If there are no restrictions, please upload the minimal anonymized data set necessary to replicate your study findings as either Supporting Information files or to a stable, public repository and provide us with the relevant URLs, DOIs, or accession numbers. For a list of acceptable repositories, please see http://journals.plos.org/plosone/s/data-availability#loc-recommended-repositories.

3. Please amend your authorship list in your manuscript file to include author Anton Kurapov,.

Reviewers' comments:

Reviewer's Responses to Questions

**Comments to the Author**

1. Is the manuscript technically sound, and do the data support the conclusions?

Reviewer #1: Partly

Reviewer #2: Partly

2. Has the statistical analysis been performed appropriately and rigorously? 

Reviewer #1: Yes

Reviewer #2: Yes

3. Have the authors made all data underlying the findings in their manuscript fully available?

Reviewer #1: No

Reviewer #2: Yes

4. Is the manuscript presented in an intelligible fashion and written in standard English?

Reviewer #1: Yes

Reviewer #2: Yes

5. Review Comments to the Author

Reviewer #1: Review of : Politicization of COVID-19 Health-Protective Behaviors in the United States: Longitudinal and Cross-National Evidence

This manuscript present two studies, the first examines the associations between political ideology, perceived COVID-19 risk and engagement in three protective behaviors in a series of US surveys. The second study expands to include mask wearing and vaccine intentions as further outcomes and compares the US against an international non-US sample.

Although the samples in these studies are large, the conclusions that can be drawn from the data are limited due to the nature of the survey roll out and structure of the data. The authors are keenly aware of this and, for the most part, clearly acknowledge these limitations.

The sample sizes are impressive and the overall statistical approach (examining correlations and mediation) is reasonable.

The basic premise of the manuscript is clear and worthwhile, essentially examining how political ideology in the US plays into the Health Belief Model (HBM) in the context of COVID-19 protective behaviors. The HBM posits that engagement in a given health behavior is the product of perceived susceptibility and severity of a disease and perceived efficacy of the behavior in preventing it (among other factors). The authors sensibly investigate the extent to which several HBM predictors mediate the established relationship between ideology and protective behavior in the US.

Below I outline my major and minor concerns with specific revisions indicated as numbered points.

Major concerns/revisions:

The biggest challenge for the studies appears to arise from the fact that political orientation was only measured at one point, and often at a different time point than the other variables in analyses. If I am incorrect about this, then a wholly different set of analyses would be more appropriate and far more informative (e.g. Random Intercepts Cross-Lagged models).

Thus the conclusions drawn are based on the assumption that political orientation is a stable individual factor that did not change throughout the survey period. I not sure how well this assumption holds, and would like to see some more discussion and justification of this. Ultimately, the authors are limited by the data they have and have made choices about how to best analyze it given those limitations.

In Study 1 the authors examine the correlations between (March baseline) political orientation and health behaviors, and identify perceived health risk of COVID-19 as significant mediator explaining the association between political orientation intended protective behvaiour across the five waves of the survey.

1) Please make clear in table 3 and 4 that is associations with ‘*baseline* political orientation’ that are presented.

In Study 2 The authors analyze the results of a large international convenience/snowball sample. After several readings I’m still not sure of the nature of surveys. A key question I have is when were people recruited? My immediate assumption was that large a sample was recruited in March and then administered follow up surveys. However, the results suggest that new participants were recruited throughout the survey period. At any given time point in Table 4, to what extent were participants returning (who had provided their political orientation in a previous wave) vs new participants (who provided their political orientation concurrently)?

2) Supplementary tables outlining the distribution of participants recruited in each wave, and their subsequent participation in following waves should be included (i.e. cross tabulating wave participation x wave recruited). The authors should also ensure that this is adequately captured in the raw data to eventually be made available with the article.

This becomes even more complicated in Table 6 where perceived risk and mask-wearing behavior were measured at different time points. This leads to a ‘cross-sectional’ mediation analysis where the effect of political orientation (as I understand it) measured at either W1,W2,W3,W4, or W5 on mask wearing at W6 is mediated by perceived risk at W5. Furthermore it is unclear how many participants are captured in such an analysis as we don’t know how many people completed both W5 and W6. It is entirely possible that I am misunderstanding this but, if so, the authors need to be clearer about *when* each construct, including political orientation was measured.

The ‘baseline’ analyses are also troublesome, in that (I assume) they cover all people who completed the baseline survey at some point between March and July. This covers a period where peoples’ perceptions and understanding of the virus would have been changing dramatically.

3) The authors should either break up the baseline analyses into separate time points, or clearly outline, both in their results and limitations, the possible problems with covering such a long time period .

Lastly the comparison between the US and non-US is a little foolhardy. Based on such an analysis, you cannot draw that conclusion that the US is somehow different to the rest of the world. It is possible that many countries are like the US in terms of politicization, and in other countries the reverse pattern plays out (i.e. liberals perceive less risk/engage in less behavior). As there is no information on the composition of the sample in terms of country (and across waves also), the reader is unable to judge.

4) Provide some indication of the extent to which other countries were represented in the data. This would not have to be a frequency table of all countries but perhaps the top 20 or so.

5) I feel it would be useful to offer some specific country comparisons (perhaps in a supplement, perhaps those where the authors have the largest sample sizes) this would offer some weight to their claim that US is different to *other countries*, rather than only comparing it to the lumped together ‘rest of the world’

6) Refrain from referring to comparing “across countries” or “between countries” or “country of residence” – non-US is not a country.

Overall, my main concerns regarding Study 2 stem from a lack of clarity about who was asked about WHAT, WHEN.

The analyses not ideal for answering the questions that the authors pose (for example HBM predictors are only considered as single mediators rather than a more comprehensive application of the full model). But I believe that they are trying to make the most of the large dataset available, covering multiple constructs in different waves. No study is perfect, and I can personally appreciate the difficulty in getting a large-scale survey off the ground in right in the middle of the first wave of an international pandemic. Given the specific time frame examined, I believe this study can make a useful contribution to the literature if appropriately revised.

Minor revisions:

Abstract

1) Remove the term ‘cross-cultural’. This was definitely not a cross-cultural analysis.

Introduction

2) P7 Capitalize protection motivation theory (for consistency with HBM)

3) P8 “…within the context of COVID-19, the group deemphasized the public health threat…” who is the group here? Conservatives? I feel like this might be referring more to conservative elites (e.g. Trump).

4) P9 there are number of citations of news articles here, which is fine. Are there any more systematic, peer-reviewed analyses of media/elite statements that could be cited to as evidence of the claim?

Study 1 Methods

5) I would be clear here, and throughout the rest of the manuscript, that what was measured was *intended* behaviors (‘I would…’) not reported behaviors (‘I have…’).

6) P15 Please report the results of your analyses controlling for demographics in a supplement.

Study 2 Methods

7) P19 “March 27th to July 13th, 2020.” Inconsistent superscripting

8) P19 “The study was approved by the Ethics…” unnecessary quote marks

9) Measuring political orientation with a 400pt scale is odd - rescaling (e.g. to -1+1) would not change the results but might make the mediation results a little more interpretable and save a few zeros.

10) P20 “Perceived Severity of Infection.” I’m pretty sure this paragraph repeats itself.

Study 2 Results

11) P23 it is great that the authors conducted multilevel analyses as a robustness checks, and I’m fine with them including the simpler analyses in the main text. But they should provide at least a summary of the results of these additional analyses in the supplementary material.

12) Again I would reiterate in this section just what is referred to when discussing ‘political orientation’ – i.e. at which point(s) it was measured.

13) Table 4, do the month rows (e.g. Late April/Early May) correspond to waves?

General Discussion

14) P31 In discussing mask use, the authors should acknowledge that the primary reason for wearing masks is to prevent the spread of the virus *to others* rather than self-protection.

15) P31 “…happening under other political leadership” – I would be specific here and note that it was democratic leadership.

16) P31 “Our studies, however, go beyond merely demonstrating that political communication has consequences that may be life-threatening.” This sentence is overstating the results, study did not investigate political communication.

17) “…allowing us to examine the stability of associations over…Ten waves in study 2” This should be rephrased – no associations were examined over all ten waves, and given political orientation was only measured once, I don’t think you can make strong claims about stability.

18) P34 “Our studies show that messages from leaders and media outlets…” again this is overstates the results. The studies did not examine media messages. ‘Indicate’ or ‘suggests’ would be more appropriate and tentative verb to use.

19) In light of the many limitations of this study, it would be good to outline a more perfect version that could be undertaken in future (e.g. examining all HBM predictors as parallel mediators of the association between politics and behavior; conducting a truly longitudinal panel study with all measures repeated at all waves, allowing statistical tests of stability and stronger casual inferences…).

Reviewer #2: In this manuscript, the [impressively large] collaboration of coauthors use two longitudinal studies of U.S. residents to show that political conservatism was

inversely associated with perceived health risk and adoption of health-protective COVID-19 behaviors over time. They also found the effects of political orientation on health-protective behaviors were mediated by perceived risk of infection, perceived severity of infection, and perceived effectiveness of the health-protective behaviors. The manuscript also includes crossnational analyses to show effects were stronger in the U.S. (N=10,923) than in an international sample (total N=51,986), highlighting the increased and overt politicization of health behaviors in the U.S.

This is an interesting study that examines aspects and implications of the relationship between political orientation and health behaviors in the case of COVID-19. It adds to a growing number of studies that have made similar observations. Although the study does not advance this literature much theoretically, it does include some additional mediating variables that contribute to our understanding of these relationships. Overall, I find the study to be generally well-written and analyzed, although I have some concerns. If these can be addressed, the study may be publishables.

First and foremost, as the authors acknowledge, the nature of the MTurk sample is problematic. The authors recognize this but dismiss the implications too readily. Why should we believe these differences did not affect results? Also, the authors should show differences between their samples (every wave) and the general US population and probe attrition in the sample further to assure readers there were no imbalances that affect results. More needs to be dine here.

The authors could also make a more compelling case by 1) presenting key patterns and findings visually in figures and 2) reporting uncertainty measures and other methodological details more clearly.

The authors also mention differences between partisanship and ideology but could do more here to distinguish and consider implications.

Finally, lots of recent work (including studies by Sander van der Linden and colleagues) on this topic is overlooked and should be integrated.

6. PLOS authors have the option to publish the peer review history of their article (what does this mean?). If published, this will include your full peer review and any attached files.

Reviewer #1: **Yes: **John Kerr

Reviewer #2: No

---

## [Author Response · Author response to Decision Letter 0]

3 Jul 2021

Reviewer 1

“The biggest challenge for the studies appears to arise from the fact that political orientation was only measured at one point, and often at a different time point than the other variables in analyses. If I am incorrect about this, then a wholly different set of analyses would be more appropriate and far more informative (e.g. Random Intercepts Cross-Lagged models).”

We considered Cross-Lagged models with random intercepts but as we only measured political orientation at Baseline, these alternative models are not useful for our data. Although we could apply them to our data, the straightforward approach of correlations and mediations allows for much easier interpretation of effect sizes and accessibility to a larger audience. Cross-lagged models would be unlikely to reveal anything different than our current analysis given the consistency of effects across time. 

“Thus the conclusions drawn are based on the assumption that political orientation is a stable individual factor that did not change throughout the survey period. I not sure how well this assumption holds, and would like to see some more discussion and justification of this. Ultimately, the authors are limited by the data they have and have made choices about how to best analyze it given those limitations.”

The reviewer is correct that our results hinge on the assumption that political orientation is a stable individual factor. Although our data cannot directly speak to this assumption, we have now included an extensive list of references that suggest political orientation in adulthood is a stable individual factor. 

“In Study 1 the authors examine the correlations between (March baseline) political orientation and health behaviors, and identify perceived health risk of COVID-19 as significant mediator explaining the association between political orientation intended protective behvaiour across the five waves of the survey.” Please make clear in table 3 and 4 that is associations with ‘*baseline* political orientation’ that are presented.”

We have updated the titles in all tables to be clear that political orientation refers to a baseline measurement. 

“In Study 2 The authors analyze the results of a large international convenience/snowball sample. After several readings I’m still not sure of the nature of surveys. A key question I have is when were people recruited? My immediate assumption was that large a sample was recruited in March and then administered follow up surveys. However, the results suggest that new participants were recruited throughout the survey period. At any given time point in Table 4, to what extent were participants returning (who had provided their political orientation in a previous wave) vs new participants (who provided their political orientation concurrently)?”

We have clarified the details of how the survey was conducted. Our goal with the study was to collect as large a cross-national sample as possible while also collecting data as the pandemic was unfolding. Most of the participants in our study (>75%) completed the Baseline survey within the first thirty days of its availability to the public. Rather than delete the participants who completed the study later—and because political orientation is a stable individual difference—we opted to retain them in the analyses. Date of Baseline survey completion was not associated with political orientation in the U.S (r = .01), but it was associated with political orientation for participations outside of the U.S. (r = .11). For this reason, we have added date of completion of the Baseline survey as a covariate to all analyses our robustness checks.

“Supplementary tables outlining the distribution of participants recruited in each wave, and their subsequent participation in following waves should be included (i.e. cross tabulating wave participation x wave recruited). The authors should also ensure that this is adequately captured in the raw data to eventually be made available with the article.”

We have added this information to the supplemental materials. Country of residence will be available in the data. 

“This becomes even more complicated in Table 6 where perceived risk and mask-wearing behavior were measured at different time points. This leads to a ‘cross-sectional’ mediation analysis where the effect of political orientation (as I understand it) measured at either W1,W2,W3,W4, or W5 on mask wearing at W6 is mediated by perceived risk at W5. Furthermore it is unclear how many participants are captured in such an analysis as we don’t know how many people completed both W5 and W6. It is entirely possible that I am misunderstanding this but, if so, the authors need to be clearer about *when* each construct, including political orientation was measured.”

We decided to simplify the analyses by averaging participants’ responses across each variable. In doing so, we represent participants’ data in a more cross-sectional way. In essence this approach represents each person’s behavior at a trait level over the available time points. This approach also resolves the problem with behaviors and beliefs being measures at different time points as all time points are now incorporated into the analysis. Given the consistency in the direction and size of our associations over time, this approach is warranted. We continue to report the original analyses separated by wave in the supplemental materials.

“The ‘baseline’ analyses are also troublesome, in that (I assume) they cover all people who completed the baseline survey at some point between March and July. This covers a period where peoples’ perceptions and understanding of the virus would have been changing dramatically. The authors should either break up the baseline analyses into separate time points, or clearly outline, both in their results and limitations, the possible problems with covering such a long time period.”

Although understanding of the virus was changing dramatically, the associations between political orientation and beliefs and behaviors related to COVID-19 did not change in our analysis over time. We have now included date of baseline survey completion as a covariate in our robustness checks; all observed interactions remain strong even after including baseline survey completion date. 

“Lastly the comparison between the US and non-US is a little foolhardy. Based on such an analysis, you cannot draw that conclusion that the US is somehow different to the rest of the world. It is possible that many countries are like the US in terms of politicization, and in other countries the reverse pattern plays out (i.e. liberals perceive less risk/engage in less behavior). As there is no information on the composition of the sample in terms of country (and across waves also), the reader is unable to judge. Provide some indication of the extent to which other countries were represented in the data. This would not have to be a frequency table of all countries but perhaps the top 20 or so. I feel it would be useful to offer some specific country comparisons (perhaps in a supplement, perhaps those where the authors have the largest sample sizes) this would offer some weight to their claim that US is different to *other countries*, rather than only comparing it to the lumped together ‘rest of the world’”

This is a fair point from the reviewer. We decided to compare the U.S. to (and report the associations within) the six countries with the largest sample at Baseline (Spain, Romania, Netherlands, Serbia, Indonesia, and Greece). Additionally, although it was not one of the top 6 countries, we thought that Canada—because it shares a border with the U.S. and might have exposure to some of the messages from U.S. politicians—would be a conservative comparison country and we included it in analyses. We compared effects across these countries in the follow-up data for consistency. We made these decisions about comparisons prior to examining any specific patterns of association in the data. We now reference these analyses in the main text and report supplemental analyses within these countries. The patterns in the U.S. differed markedly from those in Spain, Romania, Netherlands, Indonesia, and Greece. The U.S. also differed in some ways from Canada and Serbia, although the directions of the effects were consistent and the magnitude of the effects was often similar. We leave discussions of these specific patterns to political scientists. We have also added a distribution of the participants represented in the top 20 countries at Baseline and in the Follow-up analyses (see Table S2). 

Refrain from referring to comparing “across countries” or “between countries” or “country of residence” – non-US is not a country.

We have made this change throughout the manuscript. 

Overall, my main concerns regarding Study 2 stem from a lack of clarity about who was asked about WHAT, WHEN.

The analyses not ideal for answering the questions that the authors pose (for example HBM predictors are only considered as single mediators rather than a more comprehensive application of the full model). But I believe that they are trying to make the most of the large dataset available, covering multiple constructs in different waves. No study is perfect, and I can personally appreciate the difficulty in getting a large-scale survey off the ground in right in the middle of the first wave of an international pandemic. Given the specific time frame examined, I believe this study can make a useful contribution to the literature if appropriately revised.

We thank this reviewer for their careful consideration of our work and the context in which it was conducted. As they note, our aims were to provide the most comprehensive coverage of reactions to the pandemic in a manner which would be easily interpretable by a large readership. 

“1) Remove the term ‘cross-cultural’. This was definitely not a cross-cultural analysis.

Introduction

2) P7 Capitalize protection motivation theory (for consistency with HBM)

3) P8 “…within the context of COVID-19, the group deemphasized the public health threat…” who is the group here? Conservatives? I feel like this might be referring more to conservative elites (e.g. Trump).

4) P9 there are number of citations of news articles here, which is fine. Are there any more systematic, peer-reviewed analyses of media/elite statements that could be cited to as evidence of the claim?

Study 1 Methods”

We have made these changes and clarifications.

“5) I would be clear here, and throughout the rest of the manuscript, that what was measured was *intended* behaviors (‘I would…’) not reported behaviors (‘I have…’).”

We apologize for this confusion. The virus mitigation behaviors were measured in present tense (e.g., To minimize my chances of getting Coronavirus, I…. wash my hands more often.) We had added the word ‘would’ by mistake. Wearing a face covering was also captured with present tense (i.e., wearing a face covering in the last week). Only vaccine intentions were captured as intentions, as the vaccine was not available at the time of the study. We have reviewed the description of our measures to be clear about how items were measured. 

“6) P15 Please report the results of your analyses controlling for demographics in a supplement.”

We now provide these analyses in the supplement. 

“Study 2 Methods

7) P19 “March 27th to July 13th, 2020.” Inconsistent superscripting

8) P19 “The study was approved by the Ethics…” unnecessary quote marks. 

We have made these changes.

“9) Measuring political orientation with a 400pt scale is odd - rescaling (e.g. to -1+1) would not change the results but might make the mediation results a little more interpretable and save a few zeros.”

We have rescaled this measure for the mediational analyses. The reviewer was correct that doing so improved the readability of the results.

“10) P20 “Perceived Severity of Infection.” I’m pretty sure this paragraph repeats itself.”

We have addressed this issue.

“Study 2 Results

11) P23 it is great that the authors conducted multilevel analyses as a robustness checks, and I’m fine with them including the simpler analyses in the main text. But they should provide at least a summary of the results of these additional analyses in the supplementary material.”

The supplemental materials now report these analyses. 

“12) Again I would reiterate in this section just what is referred to when discussing ‘political orientation’ – i.e. at which point(s) it was measured.”

We have added reminders that political orientation was assessed only at Baseline. 

“13) Table 4, do the month rows (e.g. Late April/Early May) correspond to waves?”

Yes, in the original manuscript submission, these time frames referred to waves. We had the intuition that attaching the results to specific time points might be interesting to readers. In the current version of the manuscript, we collapse across the waves to create variables that reflect trait-like variables for each behavior for each person. As a result of this change, this particular issue is no longer relevant. We do still reference waves in the supplemental materials where we break down the results separately across time point. 

“General Discussion

14) P31 In discussing mask use, the authors should acknowledge that the primary reason for wearing masks is to prevent the spread of the virus *to others* rather than self-protection.”

We acknowledge that mask-wearing has a benefit of protecting others, and that this message was communicated by the WHO and the CDC. However, mask wearing was also communicated as a way to protect oneself. To the extent that people were more concerned about spreading COVID-19 to others than about getting it themselves, they may also have intended to become vaccinated (to reduce their chances of spreading the virus), washed their hands (to reduce spreading the virus by touching surfaces with dirty hands), avoided large crowds (so as to avoid spreading the virus), and self-quarantined if they were sick. Thus, all the health measures we assessed could measure both self- and other-protection motivations, and in this case, we were focused on the plausibility of variables related to primarily self-protection motivations. To the extent that factors such as age may have driven self- (vs. other-) motivations for engaging in the health behaviors, our robustness checks would have captured this variance. 

“15) P31 “…happening under other political leadership” – I would be specific here and note that it was democratic leadership.”

We have made this change. 

“16) P31 “Our studies, however, go beyond merely demonstrating that political communication has consequences that may be life-threatening.” This sentence is overstating the results, study did not investigate political communication.”

We have acknowledged that we do not directly test political communication and altered this paragraph. 

“17) “…allowing us to examine the stability of associations over…Ten waves in study 2” This should be rephrased – no associations were examined over all ten waves, and given political orientation was only measured once, I don’t think you can make strong claims about stability.”

We have changed the language here to avoid talking about stability of findings. 

“18) P34 “Our studies show that messages from leaders and media outlets…” again this is overstates the results. The studies did not examine media messages. ‘Indicate’ or ‘suggests’ would be more appropriate and tentative verb to use.”

We have re-written the last paragraph to be more speculative and integrate our findings with other research.

“19) In light of the many limitations of this study, it would be good to outline a more perfect version that could be undertaken in future (e.g. examining all HBM predictors as parallel mediators of the association between politics and behavior; conducting a truly longitudinal panel study with all measures repeated at all waves, allowing statistical tests of stability and stronger casual inferences…).”

We have added these points to the discussion. 

“Reviewer #2: In this manuscript, the [impressively large] collaboration of coauthors use two longitudinal studies of U.S. residents to show that political conservatism was inversely associated with perceived health risk and adoption of health-protective COVID-19 behaviors over time. They also found the effects of political orientation on health-protective behaviors were mediated by perceived risk of infection, perceived severity of infection, and perceived effectiveness of the health-protective behaviors. The manuscript also includes crossnational analyses to show effects were stronger in the U.S. (N=10,923) than in an international sample (total N=51,986), highlighting the increased and overt politicization of health behaviors in the U.S.

This is an interesting study that examines aspects and implications of the relationship between political orientation and health behaviors in the case of COVID-19. It adds to a growing number of studies that have made similar observations. Although the study does not advance this literature much theoretically, it does include some additional mediating variables that contribute to our understanding of these relationships. Overall, I find the study to be generally well-written and analyzed, although I have some concerns. If these can be addressed, the study may be publishables.

First and foremost, as the authors acknowledge, the nature of the MTurk sample is problematic. The authors recognize this but dismiss the implications too readily. Why should we believe these differences did not affect results? Also, the authors should show differences between their samples (every wave) and the general US population and probe attrition in the sample further to assure readers there were no imbalances that affect results. More needs to be done here.”

We have done extensive additional reporting of participant information by wave (see the supplemental materials). We also now more clearly describe how participants in Study 2 were collected. Although we had not highlighted this feature in the initial submission, a large subset (~25,000) of the participants in Study 2 were age and gender representative samples All participants—both paid (representative samples) and unpaid (convenience samples) were invited to participate in unpaid follow-up surveys. Thus, some of the responses in Follow-up are likely to come from participants recruited specifically to be representative. 

One of our analytic decisions in the revision also addresses the concern about attrition. That is, by focusing on the follow-up data using averages across wave for each participant, we reduce concerns about attrition—participants who responded in only one wave are included in the analysis. As Table S3 shows, although there was a very slight leftward shift among participants who participated in follow-up surveys compared to those who participated in only the Baseline survey, this shift was small and unlikely to influence the overall pattern of results. 

“The authors could also make a more compelling case by 1) presenting key patterns and findings visually in figures and 2) reporting uncertainty measures and other methodological details more clearly.”

We have reviewed our measures and expanded detail. Although we did not opt to present the findings using figures, we hope the Editor and Reviewer will see that presenting our results for the Follow-up analyses using averages does make it easier for the reader to see the consistency of patterns across variables. Note that our supplemental materials continue to present the analysis separated by wave for the reader interested in those details. 

“The authors also mention differences between partisanship and ideology but could do more here to distinguish and consider implications.”

The primary goal of this work was to connect available data on political orientation to variables in the Health Belief Model expected to relate to COVID-19 health-protective behaviors. Because this is not a political science paper, our data do not allow us to disentangle partisanship from ideology, nor to speak more to this issue than the paper currently does. 

“Finally, lots of recent work (including studies by Sander van der Linden and colleagues) on this topic is overlooked and should be integrated.”

We have extended our discussion of recent work on this topic. 

General Editorial Comments

Finally, as requested, we have made style adjustments to the documents, included Anton Kurapov to the author list, included captions, and updated file names (Tables, Figures, Supplemental Materials). 

The one lingering issue is that we are not currently in a position to place the data in a repository. We are not currently allowed to share the data from other dataset because the ethical board governing the study has deemed political orientation as a special category of personal data. As a project, we have been committed to working to get our full dataset for the larger PsyCorona study (Study 2 in the current paper) fully available for public use. This will require navigating with our data protection, data privacy, legal advisors, data management, and data security mangers at the governing institution (University of Groningen). Although we are committed to making the data available, we are not yet in a position to do so. We have enhanced transparency by linking to a codebook with exact wording (and transcriptions) of every item included in the survey. As the broader issues related to data security are resolved, we hope the data for this project will become available. At this time, however, we are under a legal and ethical obligation not to place the data in a public repository.

---

## [Decision Letter · Decision Letter 1]

23 Jul 2021

PONE-D-21-14147R1

Politicization of COVID-19 Health-Protective Behaviors in the United States: Longitudinal and Cross-National Evidence

PLOS ONE

Dear Dr. vanDellen,

Thank you for submitting your manuscript to PLOS ONE. After careful consideration, we feel that it has merit but does not fully meet PLOS ONE’s publication criteria as it currently stands. Therefore, we invite you to submit a revised version of the manuscript that addresses the points raised during the review process.

We look forward to receiving your revised manuscript.

Kind regards,

Amitava Mukherjee, ME, Ph.D.

Academic Editor

PLOS ONE

Journal Requirements:

Reviewers' comments:

Reviewer's Responses to Questions

**Comments to the Author**

1. If the authors have adequately addressed your comments raised in a previous round of review and you feel that this manuscript is now acceptable for publication, you may indicate that here to bypass the “Comments to the Author” section, enter your conflict of interest statement in the “Confidential to Editor” section, and submit your "Accept" recommendation.

Reviewer #1: (No Response)

Reviewer #2: All comments have been addressed

2. Is the manuscript technically sound, and do the data support the conclusions?

Reviewer #1: Yes

Reviewer #2: Yes

3. Has the statistical analysis been performed appropriately and rigorously? 

Reviewer #1: Yes

Reviewer #2: Yes

4. Have the authors made all data underlying the findings in their manuscript fully available?

Reviewer #1: No

Reviewer #2: No

5. Is the manuscript presented in an intelligible fashion and written in standard English?

Reviewer #1: Yes

Reviewer #2: Yes

6. Review Comments to the Author

Reviewer #1: I would like to first acknowledge the authors efforts to address my many points and their attempt to simplify the results.

Their revised analyses are now much more streamlined, but they have sacrificed some nuance in the process. I am relatively sceptical of their approach of averaging responses across waves. I’m willing to accept the proposed stability of political orientation over time. But there are inherent assumptions in averaging outcomes and mediators across waves. Fortunately the authors mitigate this to some extent by retaining the original analyses in the supplementary material, so the motivated reader can examine the effects by wave more closely and see they are relatively consistent over time. This is not the approach I would have taken, but the authors are clear about the choices they have made, and I believe their conclusions can be drawn from the results they report, so I’m willing to (begrudgingly) accept this approach.

I believe the manuscript to be publishable but note the following minor points for further revision.

I made a previous recommendation that the authors outline how future research might circumvent some of the limitations of the current study. Although their response states that they have added this to the discussion I was not able to spot it (apologies if I have missed it).

I think there may be an error in the column labels of Table S4. It currently appears that the waves were alternatingly conducted in US and non-US samples, with the table values being separated by location by row as well. I would guess that the US/Non-Us labelling in the columns is an error?

Other than that point of confusion, I found this table to be very informative for understanding the structure of dataset. I would recommend that this be included in the main text – especially as the average column represents the final variables used in the main text analyses. It would help the reader to understand what is going on a bit better. If the authors are hesitant about the size of the table then perhaps a very simple table or diagram outlining which variables were included in which waves could suffice?

I would also clarify in the main text that averages were calculated for any participant that provided a response for a given variable at least one relevant wave. That is, a given participants average score on, say, perceived efficacy of face coverings could be the average of between 1 and 4 responses, depending on how many of the relevant waves they participated in. At least that is how I interpret the revised approach – if I’m wrong then then the nature of how averages were calculated should still be clarified.

A refence on page 10 (Franz and Dhanani (2021)) is incorrectly formatted.

Table 4 – clarify the nature of the comparison column, I assume this is the F result and effect size for the PO*location interaction term in the GLM. But it is not clear in the table or text.

In Table 5 I assume the indirect effect CI is based on bootstrapping, please note in the table note or text the number of samples. I would also recommend reiterating that the PO variable was standardised in the table note.

Reviewer #2: The revisions have strengthened the manuscript and addressed my major concerns and those of the other reviewer. I leave it to the editors to render a decision about the availability of the data and any lingering concerns this may pose with respect to journal policy. In any case, the revised manuscript is theoretically and technically sound and makes a valuable contribution to the burgeoning literature onnthis topic. I support publication in its current form.

7. PLOS authors have the option to publish the peer review history of their article (what does this mean?). If published, this will include your full peer review and any attached files.

Reviewer #1: **Yes: **John R Kerr

Reviewer #2: No

---

## [Author Response · Author response to Decision Letter 1]

11 Aug 2021

Dr. Mukherjee,

We continue to appreciate you and your team of reviewers. We appreciate that all parties have acknowledged the improvements in the manuscript, many of which were made possible by the reviewers’ initial thoughtful reviews We have made the requested minor changes to the manuscript. We detail our responses to the reviewer comments below. We provide reviewer comments in bold italics and our responses below each comment.

Sincerely, 

Michelle vanDellen

Reviewer 1

“I would like to first acknowledge the authors efforts to address my many points and their attempt to simplify the results.

Their revised analyses are now much more streamlined, but they have sacrificed some nuance in the process. I am relatively sceptical of their approach of averaging responses across waves. I’m willing to accept the proposed stability of political orientation over time. But there are inherent assumptions in averaging outcomes and mediators across waves. Fortunately the authors mitigate this to some extent by retaining the original analyses in the supplementary material, so the motivated reader can examine the effects by wave more closely and see they are relatively consistent over time. This is not the approach I would have taken, but the authors are clear about the choices they have made, and I believe their conclusions can be drawn from the results they report, so I’m willing to (begrudgingly) accept this approach.”

We appreciate the reviewer’s candor. We carefully considered many possibilities for presenting our results, especially given their longitudinal consistency. The choice we ultimately made would only have been possible given this consistency and, as the reviewer notes, we were able to use supplemental materials to report the full details of each analysis to allow readers access to this information. 

“I believe the manuscript to be publishable but note the following minor points for further revision.

I made a previous recommendation that the authors outline how future research might circumvent some of the limitations of the current study. Although their response states that they have added this to the discussion I was not able to spot it (apologies if I have missed it).”

Although we had added to the discussion section statements that addressed our limitations, we did not explicitly state what we would have done in future studies. We have now added these explicit statements. 

“I think there may be an error in the column labels of Table S4. It currently appears that the waves were alternatingly conducted in US and non-US samples, with the table values being separated by location by row as well. I would guess that the US/Non-Us labelling in the columns is an error?”

Thank you for catching this mistake. That row was relevant to the Table we adapted to create Table S4. We have removed it and hope the information is now more helpful. 

“Other than that point of confusion, I found this table to be very informative for understanding the structure of dataset. I would recommend that this be included in the main text – especially as the average column represents the final variables used in the main text analyses. It would help the reader to understand what is going on a bit better. If the authors are hesitant about the size of the table then perhaps a very simple table or diagram outlining which variables were included in which waves could suffice?”

We are glad this table was helpful and we moved it to the main text as the reviewer suggested. 

“I would also clarify in the main text that averages were calculated for any participant that provided a response for a given variable at least one relevant wave. That is, a given participants average score on, say, perceived efficacy of face coverings could be the average of between 1 and 4 responses, depending on how many of the relevant waves they participated in. At least that is how I interpret the revised approach – if I’m wrong then then the nature of how averages were calculated should still be clarified.”

The reviewer’s interpretation of how averages were calculated is correct; we have added text to the document to increase this clarity. 

“A refence on page 10 (Franz and Dhanani (2021)) is incorrectly formatted.”

Because this reference was not an endote (i.e., we referred to the study as the subject of the sentence), reference formatting was not relevant. We have modified the sentence to make the reference consistent with others in the text. 

“Table 4 – clarify the nature of the comparison column, I assume this is the F result and effect size for the PO*location interaction term in the GLM. But it is not clear in the table or text.”

Table 4 is now included as Table 5. We have added text to clarify what the F and eta squared reports represent. 

“In Table 5 I assume the indirect effect CI is based on bootstrapping, please note in the table note or text the number of samples. I would also recommend reiterating that the PO variable was standardised in the table note.”

Table 5 is now Table 6. The bootstrapping approach and number of samples is already reported (see Page 29). We have added a reminder that the political orientation variable was standardized. 

“Reviewer #2: The revisions have strengthened the manuscript and addressed my major concerns and those of the other reviewer. I leave it to the editors to render a decision about the availability of the data and any lingering concerns this may pose with respect to journal policy. In any case, the revised manuscript is theoretically and technically sound and makes a valuable contribution to the burgeoning literature on this topic. I support publication in its current form.”

We thank this reviewer for their time in reviewing the manuscript.

---

## [Editor Report · Decision Letter 2]

16 Aug 2021

Politicization of COVID-19 Health-Protective Behaviors in the United States: Longitudinal and Cross-National Evidence

PONE-D-21-14147R2

Dear Dr. vanDellen,

We’re pleased to inform you that your manuscript has been judged scientifically suitable for publication and will be formally accepted for publication once it meets all outstanding technical requirements.

Kind regards,

Amitava Mukherjee, ME, Ph.D.

Academic Editor

PLOS ONE
---

## [Editor Report · Acceptance letter]

27 Sep 2021

PONE-D-21-14147R2 

Politicization of COVID-19 Health-Protective Behaviors in the United States: Longitudinal and Cross-National Evidence 

Dear Dr. vanDellen:

I'm pleased to inform you that your manuscript has been deemed suitable for publication in PLOS ONE. Congratulations! Your manuscript is now with our production department. 

Kind regards, 

on behalf of

Professor Dr. Amitava Mukherjee 

Academic Editor

PLOS ONE